# Expression of β-globin by cancer cells promotes cell survival during blood-borne dissemination

Yu Zheng[1,2,*], David T. Miyamoto[1,3,*], Ben S. Wittner[1,4], James P. Sullivan[1,†], Nicola Aceto[1,†], Nicole Vincent Jordan[1,2], Min Yu[1,†], Nezihi Murat Karabacak[5], Valentine Comaills[1], Robert Morris[1], Rushil Desai[1], Niyati Desai[1,6], Erin Emmons[1], John D. Milner[1], Richard J. Lee[1,4], Chin-Lee Wu[1,6], Lecia V. Sequist[1,4], Wilhelm Haas[1,4], David T. Ting[1,4], Mehmet Toner[5], Sridhar Ramaswamy[1,4], Shyamala Maheswaran[1,7] & Daniel A. Haber[1,2,4]

Metastasis-competent circulating tumour cells (CTCs) experience oxidative stress in the bloodstream, but their survival mechanisms are not well defined. Here, comparing single-cell RNA-Seq profiles of CTCs from breast, prostate and lung cancers, we observe consistent induction of β-globin (HBB), but not its partner α-globin (HBA). The tumour-specific origin of HBB is confirmed by sequence polymorphisms within human xenograft-derived CTCs in mouse models. Increased intracellular reactive oxygen species (ROS) in cultured breast CTCs triggers HBB induction, mediated through the transcriptional regulator KLF4. Depletion of HBB in CTC-derived cultures has minimal effects on primary tumour growth, but it greatly increases apoptosis following ROS exposure, and dramatically reduces CTC-derived lung metastases. These effects are reversed by the anti-oxidant N-Acetyl Cysteine. Conversely, overexpression of HBB is sufficient to suppress intracellular ROS within CTCs. Altogether, these observations suggest that β-globin is selectively deregulated in cancer cells, mediating a cytoprotective effect during blood-borne metastasis.

[1] Massachusetts General Cancer Center, Massachusetts General Hospital, Harvard Medical School, Charlestown, Massachusetts 02129, USA. [2] Howard Hughes Medical Institute, Chevy Chase, Maryland 20815, USA. [3] Department of Radiation Oncology, Massachusetts General Hospital, Harvard Medical School, Charlestown, Massachusetts 02129, USA. [4] Department of Medicine, Massachusetts General Hospital, Harvard Medical School, Charlestown, Massachusetts 02129, USA. [5] Center for Bioengineering in Medicine, Massachusetts General Hospital, Harvard Medical School, Charlestown, Massachusetts 02129, USA. [6] Department of Pathology, Massachusetts General Hospital, Harvard Medical School, Charlestown, Massachusetts 02129, USA. [7] Department of Surgery, Massachusetts General Hospital, Harvard Medical School, Charlestown, Massachusetts 02129, USA. * These authors contributed equally to this work. † Present addresses: Google Life Sciences, Mountain View, California 94043, USA (J.P.S.); University of Basel, Basel 4058, Switzerland (N.A.); University of Southern California, Los Angeles, California 90033, USA (M.Y.). Correspondence and requests for materials should be addressed to S.M. (email: smaheswaran@mgh.harvard.edu) or to D.A.H. (email: dhaber@mgh.harvard.edu).

Cancer metastasis is an inefficient process, with only a small proportion of tumour cells successfully surviving dissemination through the bloodstream to colonize distant sites[1]. Among the recognized challenges faced by these metastatic precursors of epithelial cancers are loss of contact with basement membrane and extracellular matrix, as well as shear stress as they circulate in the vasculature[2–4]. These stresses may induce apoptotic signals such as anoikis, or cause physical damage to cell structures. Oxidative stress associated with increased intracellular ROS levels is also linked to loss of matrix adhesion and nutrient deprivation. Under these conditions, skewed redox balance can be restored by oncogenic signalling or stress signalling, which prevents the accumulation of excessive ROS within cells and prolongs cell survival[3,5]. Furthermore, antioxidants have recently been shown to accelerate lung cancer progression and melanoma metastasis in mouse models[6,7]. A potential link between antioxidants and cancer risk has also been suggested in multiple clinical studies[8–10].

Recently developed technologies to isolate circulating tumour cells (CTCs) within the vasculature provide an opportunity to dissect this transient but critical state in the metastatic process[11–13]. CTCs are extremely rare, even in patients with advanced cancer (estimated at one CTC in a billion normal blood cells). Among the diverse technologies developed to capture CTCs from blood specimens, microfluidic devices provide the advantage of high-efficiency and gentle cell handling of unfixed and unprocessed blood, thereby maximizing RNA quality for expression profiling[14]. The recently developed CTC-iChip has the added advantage of magnetically depleting normal leucocytes away from untagged CTCs, and enriching CTCs in solution while avoiding the inherent bias in using epithelial markers such as EPCAM to select a subset of tumour cells within blood specimens[15]. Most cancer cells isolated by microfluidic technologies are single cells, although a subset are captured as clusters of tumour cells (so-called CTC-clusters) ranging from 2 to >10 cells tethered together as they circulate in the bloodstream[13]. In recent studies, we performed single-cell RNA sequencing of CTCs isolated from patients with cancers of the prostate, breast and pancreas, and identified increased expression of Wnt signalling pathways, intercellular adhesion molecules and extracellular matrix components in subsets of these cells[4,13,16]. The use of single-cell sequencing allows for discrimination among heterogeneous subpopulations of CTCs, and ensures against contamination by normal blood cells. The high quality of RNA extracted from CTCs freshly isolated using negative depletion microfluidic technology is critical to enabling genome-wide single-cell RNA sequencing analyses[4,13,16].

By comparing single-cell transcriptome profiles of CTCs from multiple different human cancers, we identify β-globin (HBB) as one of the transcripts most consistently overexpressed in these tumour-derived cells. Using mouse xenograft models, we confirm the tumour cell-derived origin of the HBB transcript, taking advantage of human/mouse sequence polymorphisms. In lung cancer cells and in cultured breast CTC lines, we show that both matrix deprivation and ROS exposure result in the KLF4-dependent induction of HBB, which suppresses ROS-mediated cytotoxicity, thereby enhancing anchorage-independent cancer cell survival and facilitating distant metastasis.

## Results

**β-globin is abundantly expressed in circulating tumour cells.** The identification of transcripts that are commonly upregulated in CTCs from different types of human cancer may reveal novel mechanisms of hematogenous metastasis. Using CTC-iChip microfluidic purification of cancer cells from the blood of patients

with metastatic cancer, we have previously established single-cell transcriptome profiles of CTCs from breast[13,17] and prostate cancers[4], and we now generate single CTC RNA profiles from non-small cell lung cancer (NSCLC) xenograft models (see Methods). Patient-derived CTCs were initially identified from microfluidic leucocyte-depleted blood samples by cell surface staining (prostate: EPCAM and CDH11; breast: EPCAM, CDH11 and ERBB2), and the absence of staining for the hematopoietic markers (CD45, CD14 and CD16). NSCLC xenograft-derived CTCs were identified by their GFP expression. The identity of single CTCs was then verified through RNA sequencing analysis, confirming the expression of epithelial makers (EPCAM, KRT7, KRT8, KRT18 or KRT19), tissue lineage-specific markers (AR, KLK3, FOLH1 for prostate; ERBB2 for breast)[4,13], and the absence of hematopoietic lineage-associated transcripts (CD45, CD14, CD16 and CD114) (Fig. 1a; Supplementary Fig. 1a). For each tumour type, we identified transcripts that are enriched in CTCs compared with either their tumour of origin (prostate, lung) or with established ATCC cancer cell lines (breast) (see Methods). Unexpectedly, a haemoglobin gene, HBB (encoding β-globin), but not its binding partner HBA (encoding α-globin), was significantly overexpressed in CTCs across all three tumour types (Fig. 1a). Expression of HBB was significantly elevated in >50% of CTCs from breast, prostate and lung cancers (Reads Per Million: $RPM_{median} = 3,630$, range (1.6–894,101); $P = 2.59E–07$ for CTCs versus primary tumour and cancer cell lines) (Fig. 1b; Supplementary Fig. 1b). Expression of HBB in CTCs was heterogeneous across different patients, as well as among different CTCs within individual patients (Supplementary Fig. 1c,d).

To ensure that the HBB transcripts are truly derived from epithelial tumour cells, rather than potentially contaminating red blood cells (RBCs), we made use of human versus mouse HBB coding sequence polymorphisms, which are readily measurable by RNA sequencing of human xenograft-derived CTCs, admixed with murine red blood cell background (Supplementary Fig. 2a). Indeed, microfluidic isolation of human tumour-derived CTCs from NSCLC xenografts followed by single-cell RNA sequencing demonstrated high levels of expression specific to the human HBB transcript ($RPM_{median} = 99$, range (4.6–678,887)) (Supplementary Figs 1b, 2b). We also note that in CTCs isolated from this xenograft model, human HBB is expressed at a discoordinately high level compared with HBA, whereas mouse RBCs and their reticulocyte precursors co-express comparable amounts of the murine α-chain of haemoglobin (HBA) and beta-chain (HBB) (Supplementary Fig. 2c). The specific overexpression of HBB, but not HBA, is also evident using single-cell RNA-Seq of CTCs and CTC-clusters from multiple independent breast and prostate cancer patients (Fig. 1c).

To further confirm the tumour origin of HBB, we used RNA in-situ hybridization (RNA-ISH) to directly visualize the cells expressing HBB. In blood samples from 5 patients with advanced prostate cancer (castrate-resistant or CRPC), 58 of 134 single CTCs and CTC-clusters that were positive for expression of EPCAM and/or KRT8/18/19 transcripts coexpressed HBB mRNA (Fig. 1d,e). CTCs undergo multiple physical and metabolic challenges as they circulate in the blood, raising the possibility that HBB induction may reflect some form of cellular stress such as oxidative stress. To test if such oxidative stress is detectable in CTCs, we used live cell staining with H2DCFDA dye to quantify intracellular ROS levels, and with MitoSOX red dye to detect physiological mitochondrial superoxide in both CTCs and white blood cells (WBCs). Mitochondrial superoxide staining was evident in 16/17 CTCs

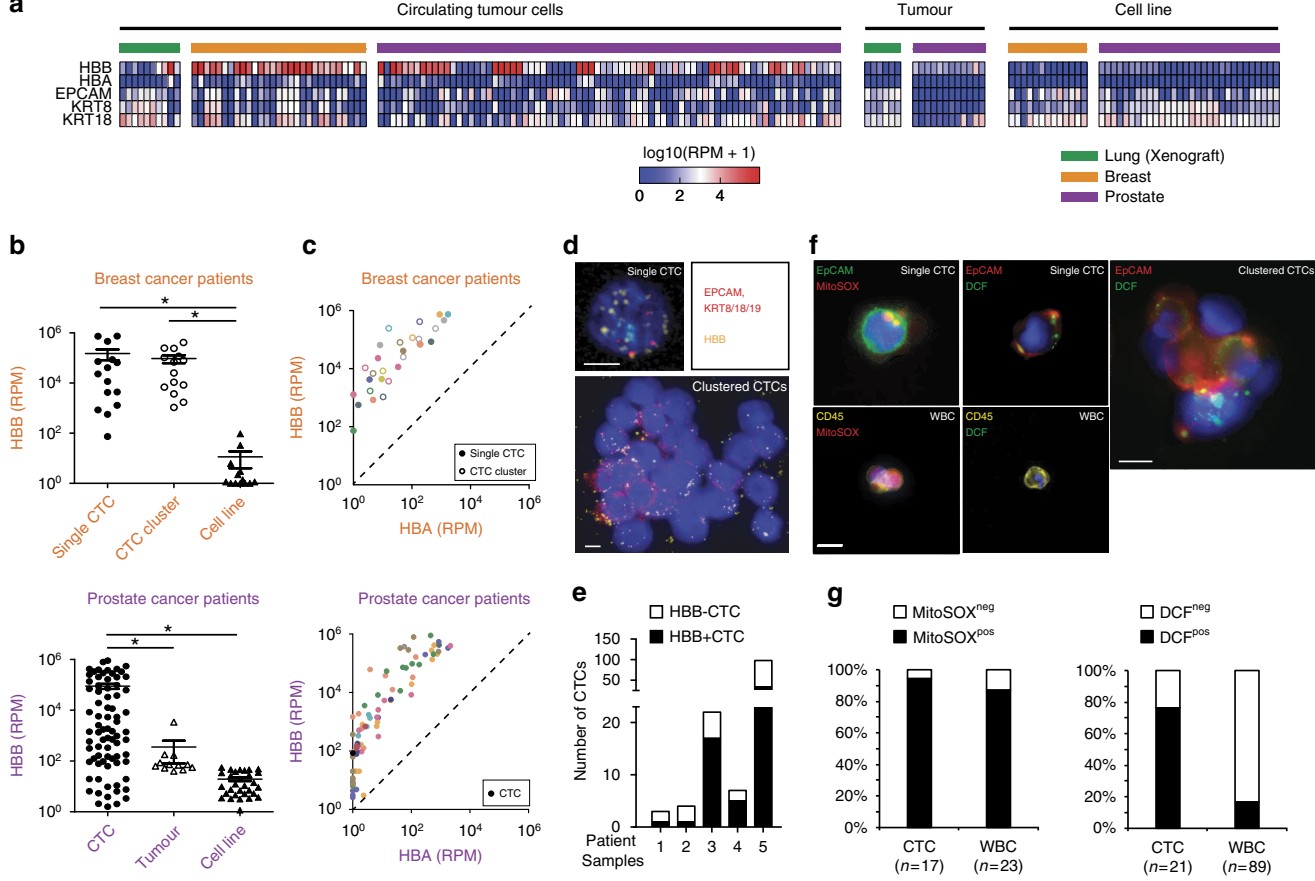

**Figure 1 | Elevated expression of *HBB* is detected in circulating tumour cells.** (**a**) Heat map showing relative expression of haemoglobin genes (*HBB* and *HBA*) and epithelial lineage genes (*EPCAM, KRT8* and *KRT18*) in single CTCs and clustered CTCs (lung, $n = 10$; breast, $n = 29$ and prostate, $n = 77$), primary tumour samples (lung, $n = 6$; prostate, $n = 12$), and established cancer cell lines (breast, $n = 13$; prostate, $n = 4$). Data are derived from single-cell RNA sequencing (see Methods). (**b**) Scatter plot showing increased expression of *HBB* in single CTCs or CTC-clusters isolated from clinical blood samples of patients with metastatic breast or prostate cancer, compared with single primary tumour cells and/or established cancer cell lines. Data are represented as mean ± s.e.m. *denotes $P < 0.05$ (t-Test). (**c**) Scatter plot demonstrating the specific upregulation of *HBB* but not *HBA* in CTCs of breast and prostate cancer patients. Each colour represents a different patient. (**d**) Representative fluorescence images ($\times 40$) of RNA-*in situ* hybridization of a single prostate CTC and a prostate CTC-cluster show expression of *HBB* (yellow dots), and the epithelial lineage markers, *EPCAM* and *KRT8/18/19*. DNA is stained with 4′, 6′-diamidino-2-phenylindole (blue). Scale bar, 10 μm. (**e**) Quantitation of RNA-*in situ* signal defining *HBB*-negative and *HBB*-positive CTCs from a cohort of 5 patients with advanced prostate cancer. (**f**) Representative fluorescence images of DCF or MitoSOX staining of a single prostate CTCs or CTC-clusters. Prostate CTCs were identified by *EpCAM* staining, and white blood cells (WBCs) were identified by *CD45* positivity. DNA is stained with DAPI. Scale bar, 10 μm. (**g**) Quantitation of fluorescent signal defining MitoSOX-positive and DCF-positive CTCs and WBCs from a cohort of 5 patients with advanced prostate cancer. The total number of cells analysed is indicated.

and 20/23 WBCs from 5 patients, consistent with cell viability. In striking contrast, 16/21 CTCs (including CTC clusters), but only 15/89 WBCs had cytosolic accumulated ROS levels ($P = 5.4\text{E}{-}08$) (Fig. 1f,g).

The elevated expression of *HBB* in cancer cells appears to be largely restricted to CTCs. Minimal or no expression of this globin chain is detected in primary lung and prostate cancers or in most ATCC breast cancer cell lines cultured under standard 2D conditions (Fig. 1a). In contrast, primary breast cancer CTC-derived cell cultures, which are maintained under anchorage-independent culture conditions, have elevated expression of *HBB* ($\text{RPM}_{median} = 56$, range (16–133)) for CTCs versus ATCC breast cancer cell lines ($\text{RPM}_{median} = 0.1$, range (0 to 94), $P = 0.0021$) (Supplementary Fig. 3a). The *in vitro* expansion of CTC-derived cell lines makes it possible to generate sufficient material for mass spectrometric analysis. These oligoclonally-derived CTC cell lines have abundant β-globin protein levels, whose expression varies across a dynamic range in different

CTC cell lines (Supplementary Fig. 3b), with a median of 0.012% of the β-globin levels present in RBCs (see methods). Consistent with the RNA analyses, no α-globin protein is detectable by mass spectrometry in cultured breast CTCs. Taken altogether, the β-chain of haemoglobin is specifically and consistently upregulated within CTCs, compared with their respective tumour cells of origin.

**Upregulation of *HBB* is mediated by *KLF4* in response to ROS.** The increased oxidative stress detected in CTCs raises the possibility that expression of the oxygen and haem binding β-globin in CTCs is linked to accumulated intracellular ROS levels. To test this, we used cultured breast CTCs (BRx50), which express elevated levels of *HBB* at baseline (Supplementary Fig. 3a,b) and proliferate best under anchorage-independent hypoxic conditions[17], as well as two other established cancer cell lines, H727 (lung carcinoid) and MGH134 (non-small cell lung

cancer)[18], both of which grow under adherent conditions and have low levels of endogenous *HBB* transcripts. Exposure to hydrogen peroxide, which increases intracellular ROS, leads to a dramatic induction of *HBB* RNA in a time-dependent manner in both BRx50 and H727 cells (Fig. 2a). In contrast, other stress-related environmental stimuli, such as short-term serum deprivation (0.1%FBS) and hypoxia (4% $O_2$), do not affect *HBB* expression. Culture of the normally attached H727 cells under anchorage-independent conditions shows a modest induction of *HBB* expression (Supplementary Fig. 4a). Of note, loss of adherence to matrix is known to be a physiological trigger of increased intracellular ROS (ref. 3) (Supplementary Fig. 4b). Non-ROS mediated interventions that impair cell–cell interactions, such as suppressing the adhesion molecules E-cadherin or Integrin B1, do not result in *HBB* induction (Supplementary Fig. 4c). Importantly, pre-treatment of BRx50 and H727 cells with the ROS scavenger N-acetyl cysteine (NAC) abolishes the upregulation of *HBB* induced by hydrogen peroxide, consistent with the dependence of this process on the accumulation of intracellular ROS (Fig. 2b). Pre-treatment of H727 cells with NAC also impairs the *HBB* induction caused by matrix deprivation (Fig. 2b). Similar observations were made using MGH134 cells (Supplementary Fig. 4d,e).

To identify candidate regulators of *HBB* induction in tumour cells, we first screened for transcription factors and chromatin regulators whose expression is positively correlated with *HBB* levels in multiple expression datasets of human cancer (see methods and Supplementary Fig. 5a). We also expanded the number of *KLF* gene family members in this screening group, because there is a canonical *KLF1* binding site (a CACCC motif) within the *HBB* promoter[19], and not all *KLF* genes are represented in the available microarray data. Out of 21 candidates, only three genes, *KLF4*, *KLF6* and *ATF5*, are induced under both hydrogen peroxide-treated and anchorage-free culture conditions (Supplementary Fig. 5b,c). Depletion of *KLF4* in BRx50 cells using two different siRNAs suppresses *HBB* under basal conditions, as well as following ROS stress, while knockdown of *ATF5* or *KLF6* shows a more modest effect (Fig. 2c; Supplementary Fig. 6a). Similar observations were made in H727 cells (Fig. 2c; Supplementary Fig. 6a). We further tested the binding of *KLF4* to its putative target motif CACCC within the *HBB* gene promoter using chromatin-immunoprecipitation (chromatin-IP). Direct binding of *KLF4* to the *HBB* gene promoter in H727 cells is evident at baseline, and this binding is rapidly enhanced following treatment of cells with hydrogen peroxide (Fig. 2d). Of note, *KLF4* is also abundantly expressed and positively correlated with *HBB* levels in a cohort of 6 patient-derived breast CTC cultures, compared with a panel of 13 established breast cancer-derived cell lines ($P = 6.6E–04$) (Supplementary Fig. 6b). Increased expression of *KLF4* is also evident in prostate CTCs compared with primary tumour specimens in our single-cell RNA-Seq data (Supplementary Fig. 6c), while its upregulation is less clear in CTCs from lung xenografts because of limited sample size in that cohort. We note that KLF1 is a known regulator of β-globin expression, but not α-globin expression in erythrocytes[19]. Hence the role of its close family member *KLF4* in selectively inducing *HBB* but not *HBA* in epithelial cells suggests an analogous regulatory pathway in response to ROS-mediated signals.

**HBB protects cancer cells from ROS-induced apoptosis**. Expression of both α- and β-globin chains by some epithelial cancers has been described, and their ectopic overexpression in some cancer cell lines have been linked to ROS resistance[20–22]. In our single CTC RNA-Seq data, the specific induction of *HBB* but not *HBA* expression suggests a role for β-globin alone in ROS quenching. To model the functional consequences of specific *HBB* modulation in cancer cells circulating in the bloodstream, we therefore performed loss-of-function experiments using shRNAs against *HBB* to test ROS sensitivity and metastatic propensity in mouse models.

Knockdown of endogenous *HBB* in cultured CTCs (BRx50) significantly suppresses their colony formation in soft agar (Fig. 3a,b, top panel), a condition where overcoming accumulated ROS is key for the survival[3]. *HBB* knockdown also reduces their proliferation rate under anchorage-independent suspension culture (Fig. 3c, top panel), an effect that is accompanied by increased apoptosis and elevated intracellular ROS levels (Fig. 3d,e, top panel). Importantly, the effect of *HBB* depletion on cell survival is rescued by pre-treating cells with the anti-oxidant NAC, confirming the contribution of ROS to the phenotype (Fig. 3d,e, top panel). Similar observations were made in adherent H727 Cells (Fig. 3a–e, bottom panel). Under standard adherent culture conditions, knockdown of *HBB* in these cells shows a modest ($P = 1.2E–04$) suppression of proliferation, which is correlated with increased apoptosis (from $1.3 \pm 0.1\%$ to $16.0 \pm 4.0\%$, $P = 0.008$), as measured by Annexin V staining and subG1 fractions (Supplementary Fig. 7a–c). However, under anchorage-independent conditions, *HBB*-depleted H727 cells demonstrate a dramatic increase in cell death (from $22.5 \pm 0.9\%$ to $44.3 \pm 2.9\%$, $P = 3.9E-05$), and they fail to form any colonies in long-term methylcellulose soft agar assays (Fig. 3b,c, bottom). As breast CTC cultures, H727 cells with *HBB* knockdown have elevated intracellular ROS levels and increased apoptosis, which can be rescued by pretreatment with NAC (Fig. 3d,e, bottom). In addition to anchorage-independent growth, both *HBB*-depleted BRx50 and H727 cells show increased sensitivity to ROS induced by exogenous hydrogen peroxide ($H_2O_2$) (Fig. 3f; Supplementary Fig. 7d). Of note, shRNA-mediated depletion of *HBB* in H727 cells does not alter intracellular oxygen levels, as measured by intracellular $O_2$ sensor probes (Supplementary Fig. 7e), nor does it affect the migration and invasive potential of H727 cells in Boyden chamber assays (Supplementary Fig. 7f,g). As a further control for shRNA target specificity, infection of prostate LNCaP cells, which lack endogenous *HBB* expression, does not affect cell proliferation (Supplementary Fig. 7h).

Complementing these knockdown experiments, ectopic expression of *HBB* in BRx50 and H727 cells is sufficient to reduce intracellular levels of ROS under basal conditions, as well as following treatment with hydrogen peroxide (Fig. 3g,h; Supplementary Fig. 7i,j). Moreover, measurements of intracellular iron content demonstrate increased iron levels in BRx50 and H727 cells overexpressing *HBB*, consistent with the hypothesis that the formation of β-globin/haem/iron complex contributes to the anti-oxidant function of *HBB*, partly through sequestering iron from radical-generating reactions (Fig. 3i; Supplementary Fig. 7k). Taken together, these data suggest that the induction of *HBB* following ROS-related stress attenuates intracellular ROS levels and contributes to the survival of epithelial cancer cells.

**HBB contributes to the metastatic potential of breast CTCs**. BRx50 cultured CTCs are tumorigenic following inoculation into immunosuppressed NSG mice[17], and therefore can be used to evaluate their metastatic potential *in vivo*. Inoculation of 200,000 BRx50 cells into a mammary fat pad produces measurable tumours within 4 weeks; simultaneous inoculation of *HBB*-depleted BRx50 cells into other mammary fat pads within the same mouse shows a modest but significant initial delay in primary tumour formation ($n = 4$; $P = 0.05$), which subsequently

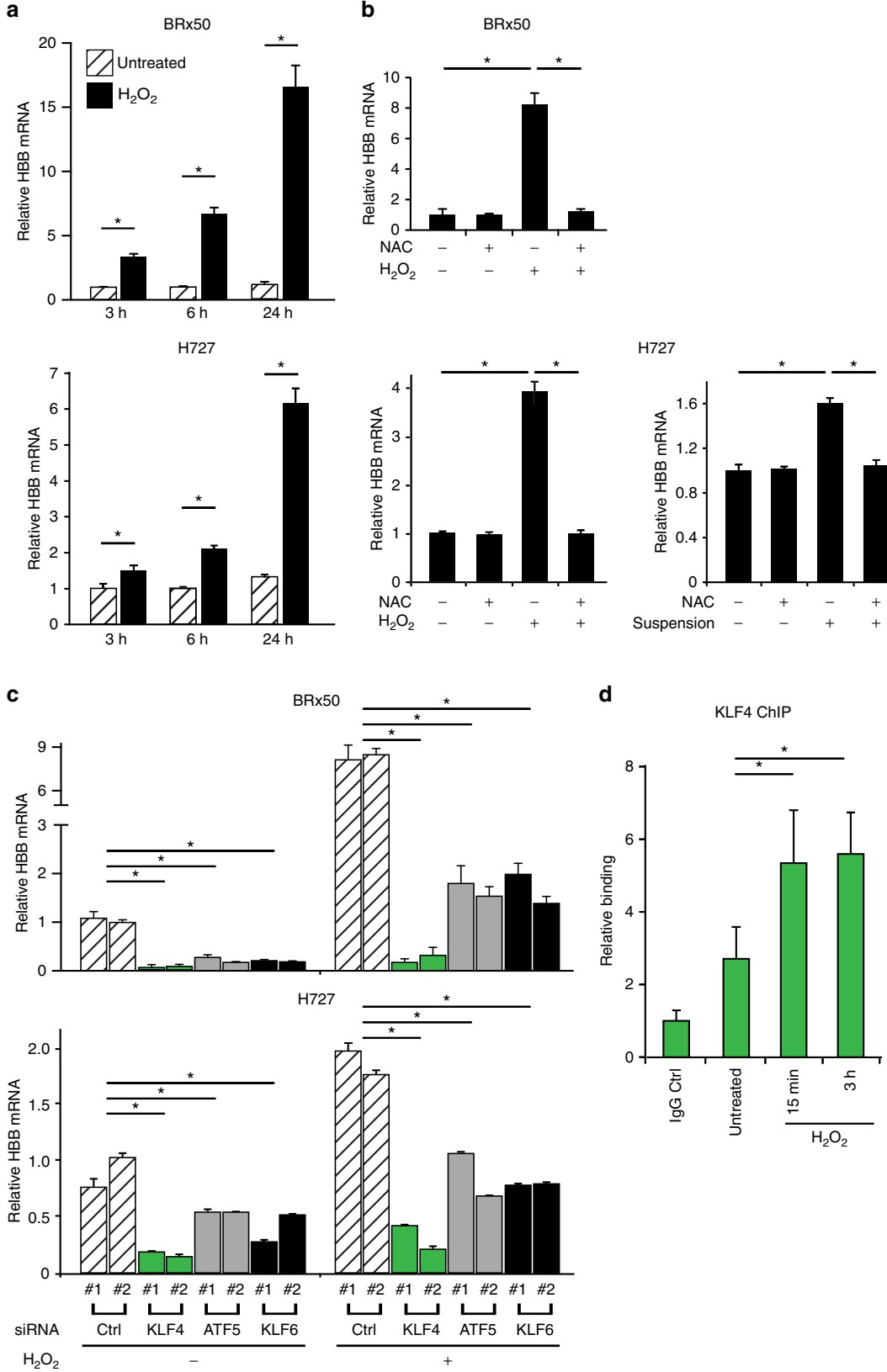

**Figure 2 | *HBB* induction is mediated through *KLF4* in response to increased intracellular ROS.** (**a**) Bar graph showing time-dependent induction of *HBB* mRNA levels in $H_2O_2$ treated BRx50 and H727 cells. (**b**) Bar graphs showing suppression of $H_2O_2$-induced *HBB* expression by the anti-oxidant N-acetyl cysteine (1 mM NAC) in BRx50 and H727 cells, and suppression of suspension-induced *HBB* expression by NAC in H727 cells. (**c**) Bar graph showing that *KLF4* depletion significantly decreases *HBB* expression under both basal and stimulated conditions in BRx50 and H727 cells. Knockdown of *ATF5* or *KLF6* has a modest effect. (**d**) ChIP assay showing direct binding of *KLF4* to the promoter of *HBB* under basal conditions, with increased binding following exposure to $H_2O_2$ in H727 cells. All data are represented as mean ± s.d. $n = 3$; *denotes a statistical significance at $P < 0.05$ (*t*-Test).

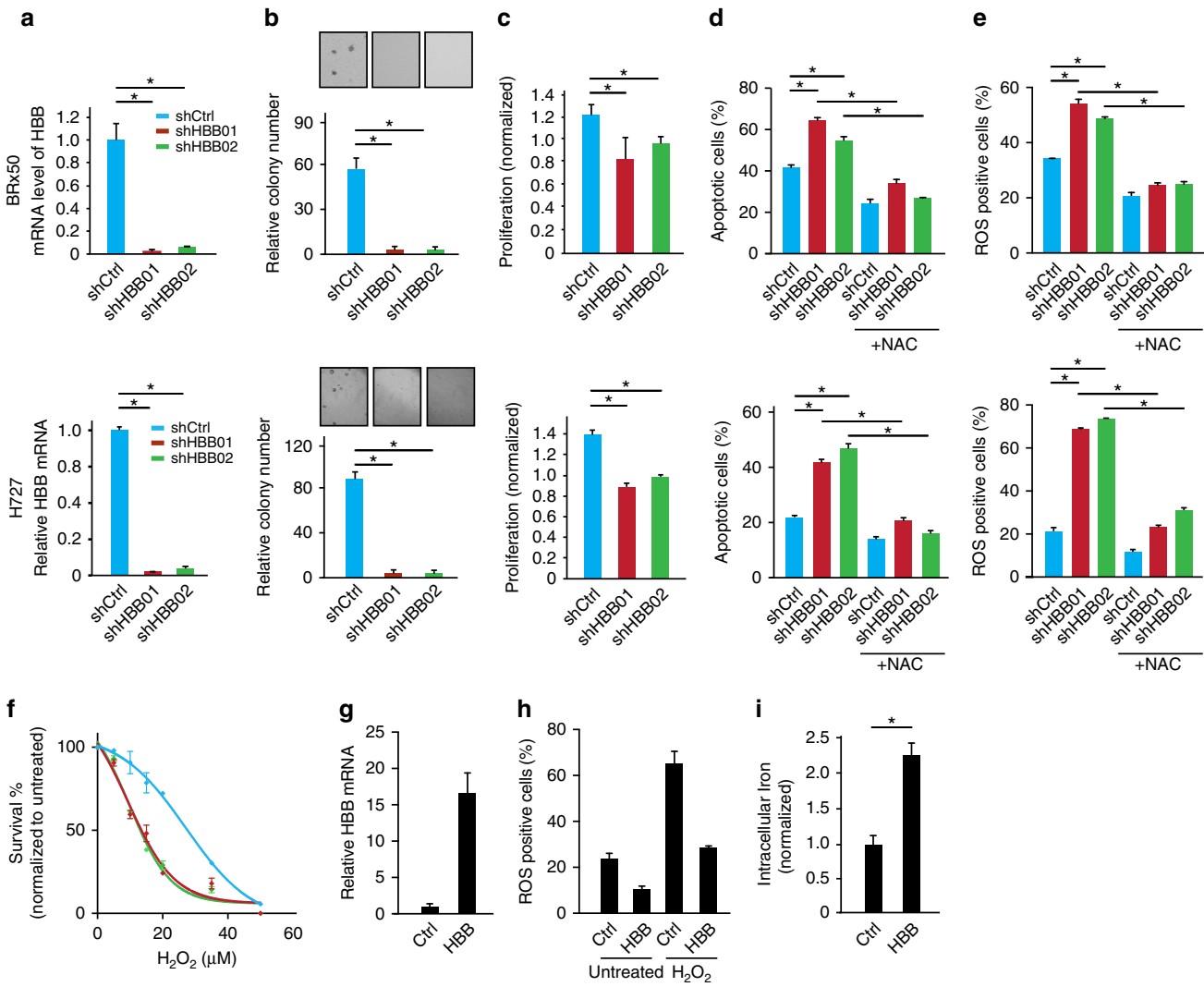

**Figure 3 | *HBB* contributes to the survival of tumour cells under ROS stress.** (**a**–**e**) top panel: BRx50 cells; bottom panel: H727 cells (**a**) Bar graph showing effective shRNA-mediated knockdown of *HBB*. (**b**) Bar graph showing that *HBB* depletion impairs colony formation in soft agar (measured at 3 weeks). Representative images are shown. (**c**) Bar graph showing that depletion of *HBB* reduces short-term proliferation of BRx50 cells (5 days) and H727 cells (24 h) cultured in suspension. (**d**,**e**) Bar graph showing that *HBB* depletion increases apoptosis and intracellular ROS levels in cells cultured in suspension for 24 h; pre-treatment of cells with the anti-oxidant NAC, rescues both ROS levels and apoptosis. (**f**) Bar graph showing that *HBB*-depleted H727 cells exhibit increased sensitivity to $H_2O_2$ compared with control cells. (**g**) Real-time PCR showing the relative *HBB* mRNA levels in BRx50 cells at baseline (Ctrl) or following stable expression of ectopic *HBB*. (**h**) Bar graph showing that ectopic overexpression of *HBB* in BRx50 cells reduces intracellular ROS under basal conditions and following treatment with hydrogen peroxide. (**i**) Bar graph showing that overexpression of *HBB* increases total iron within BRx50 cells compared with control cells. All data (**a**–**i**) are represented as mean ± s.d. $n = 3$; *denotes a statistical significance at $P < 0.05$ (*t*-Test).

resolves, leading to primary tumours of comparable size within 8 weeks (Fig. 4a; Supplementary Fig. 7l). In contrast to this modest effect on primary orthotopic tumour generation, we observed a dramatic effect on metastatic potential following tail vein injection. Direct intravascular introduction of 50,000 cultured BRx50 CTCs into their origin in the bloodstream results in multiple lung metastases in NSG mice, whereas *HBB*-depleted CTCs show significantly impaired potential in generating metastatic lesions (Fig. 4b; Supplementary Fig. 8a,b). A similar abrogation of tail vein-initiated lung metastases following knockdown of endogenous *HBB* expression is also evident using H727 cells (Supplementary Fig. 8c). To test whether *HBB* depletion induced loss of metastatic potential is because of accumulated intracellular ROS levels in circulating tumour cells, we pre-treated mice with the anti-oxidant NAC for 3 days, and then inoculated BRx50 cells with either

*HBB* knockdown or control constructs via tail vein injection. Consistent with a critical role for oxidative stress in cancer cell survival in the bloodstream, NAC pretreatment of mice promotes lung tumorigenesis by intravascularly injected CTCs, and it significantly rescues the metastatic potential of these cells following *HBB* knockdown (Fig. 4c).

**Roles of *HBB* and other anti-oxidants in CTCs and metastasis.** Extending our observations based on mouse models of metastasis to clinical outcome data, we performed a meta-analysis of *HBB* expression using multiple publicly available expression datasets containing both primary and metastatic breast cancer samples. The composite meta-score demonstrates elevated expression of *HBB* in metastatic tumours compared with primary tumours ($P = 0.019$), with 6 out of 19 datasets exhibiting

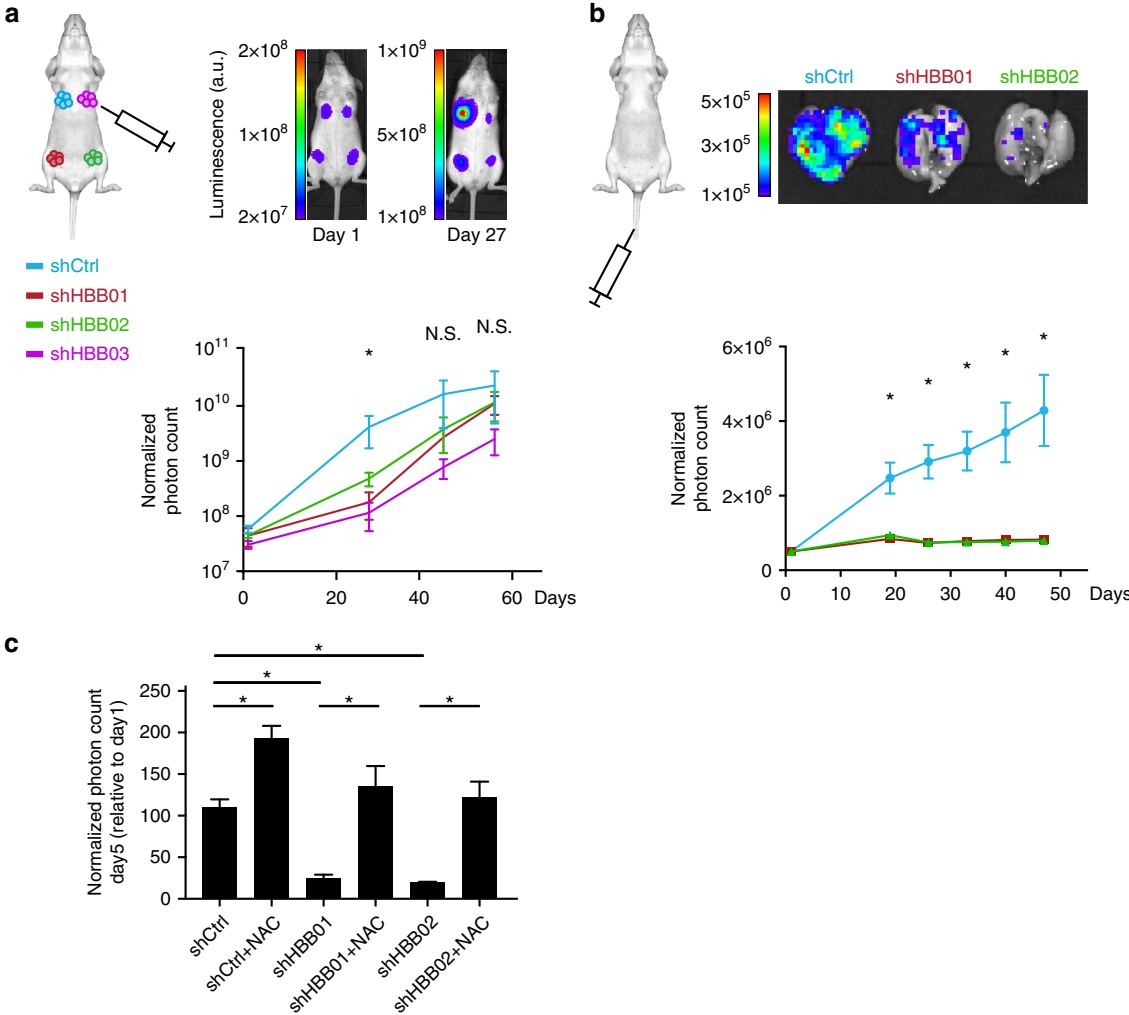

**Figure 4 | *HBB* expression enhances the ability of breast CTCs to form distant metastases.** (**a**) Orthotopic inoculation of 200,000 BRx50 CTCs into the mammary fat pad showing transiently delayed primary tumour formation upon *HBB*-depletion. *Top*: representative images of mice at day 1 and day 27 following tumour inoculation (luminescence by IVIS imaging). For each mouse, the mammary fat pad seen in the upper left of the image received control CTCs, while the other three fat pads received CTCs expressing three different shRNA constructs targeting *HBB*. Bottom: Primary tumour growth curves from mice orthotopically implanted with BRx50 CTCs expressing either control or *HBB* shRNAs. The transient but significant reduction in tumour size at day 27 is resolved in subsequent time points. (**b**) Intravenous injection of BRx50 cultured CTCs into the mouse tail vein, showing impaired metastatic potential following depletion of endogenous *HBB*. Top: representative IVIS-luminescence images of isolated whole lungs from mice, at day 33 following intravenous injection of CTCs expressing either control or *HBB*-targeting shRNAs. Bottom: longitudinal monitoring of mice for emergence of lung metastases, following intravascular injection of either control BRx50 cells or BRx50 cells expressing *HBB* shRNAs. (**c**) Bar graph showing normalized lung photon counts from mice, representing metastatic tumour burden, 5 days after they were intravenously injected with either control BRx50 cells or *HBB*-depleted BRx50 cells. Where indicated, mice were pretreated with the antioxidant NAC ($200\,\mathrm{mg\,kg^{-1}day^{-1}}$) for 3 days before tumour cell inoculation, and then treated daily at the same dose. Total lung photon counts at day 5 were normalized to the photon counts at day 1. Data (**a**–**c**) are represented as mean ± s.e.m.; $n = 4$ in (a), $n = 3$ in **b**,**c**; *denotes a statistical significance at $P < 0.05$ (*t*-Test).

a positive score with a significant *P* value (Fig. 5a). Indeed, *HBB* mRNA level in primary tumour samples is generally low (0–0.9%), while increased expression is detected in 7 out of 19 metastatic prostate tumours (6.3–18.1%) ($P = 0.042$) (Fig. 5b; Supplementary Fig. 9). While *HBB* expression is comparably low in primary tumours, we used multiple primary datasets to test whether any baseline *HBB* expression in primary tumours from multiple tissue types is correlated with patient outcome. Kaplan–Meier survival analyses in several datasets show *HBB* expression to be a significant predictor for a poor clinical outcome (Supplementary Fig. 10a–c; Supplementary Table 1).

To obtain a full view of the change of *HBB* expression in the progression from primary tumour to CTCs and ultimately distant metastasis, we took advantage of a prostate cancer mouse model[23], in which we obtained single-cell transcriptome profiles of tumour cells at each of these different stages. Expression of *HBB* is low in primary tumour cells, markedly induced in CTCs, and then again reduced in metastases (Supplementary Fig. 11a,b). It is noteworthy that substantial heterogeneity in *Hbb* expression is observed in single tumour cells isolated from metastases, with 8 out of 22 cells showing substantial expression of *Hbb* ($\mathrm{RPM_{median}} = 65$, range (23–1,858)), while the rest have minimal or no expression (Supplementary Fig. 11a). While such simultaneous comparisons of primary tumour, CTCs and metastatic tumours are challenging in clinical cohorts, we tested for a correlation between *HBB* expression in CTCs and tumour

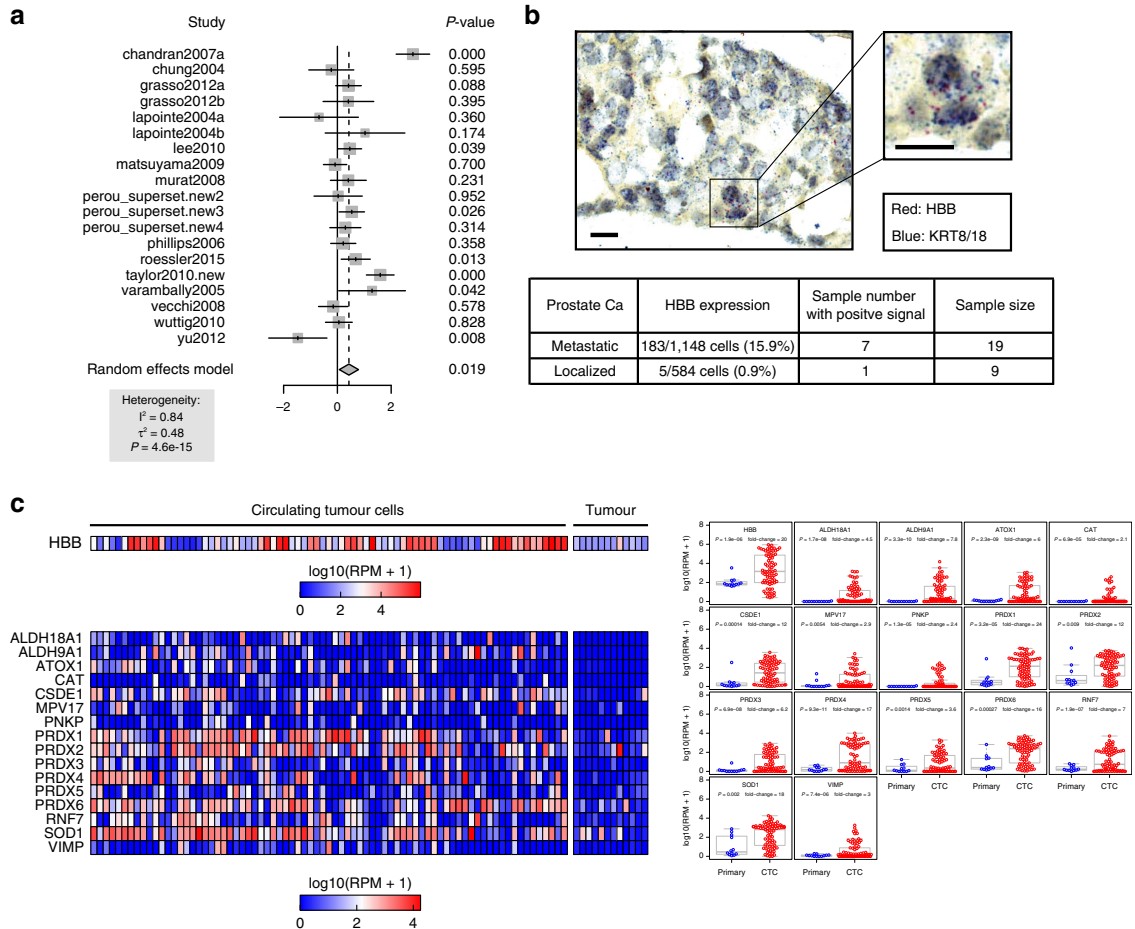

**Figure 5 | Roles of *HBB* and other anti-oxidants in CTCs and metastasis.** (**a**) Meta-analysis of *HBB* expression using multiple publicly available expression datasets showing increased expression of *HBB* in different metastatic cancers, compared with primary tumours ($P = 0.019$, see Methods). (**b**) Representative micrograph ($\times 40$) of RNA-*in situ* hybridization (multimerized oligoprobes, Affymetrix) of a metastatic prostate tumour showing the expression of *HBB* (red) and the epithelial keratin markers *KRT8/18* (blue). The table summarizes the level of *HBB* expression and the number of primary and metastatic prostate cancer specimens examined. Scale bar, 20 µm. (**c**) Heat map and box plots showing upregulated expression of *HBB* and 16 other antioxidant genes in prostate CTCs versus primary tumour. The *HBB* heatmap uses a different colour scale, since it has a significantly higher number of reads, compared with the other genes. Fold changes and *P* values for *HBB* and the overexpressed antioxidant genes are indicated in the box plots.

progression in a cohort of patients with advanced castration-resistant prostate cancer (CRPC). In four clinical cases for which longitudinal follow-up was available, we observed a significant increase of *HBB* expression by single-cell RNA sequencing of individual CTCs in patients who have radiographic evidence of progression, compared with those who have stable disease ($P = 3.2E-06$, Supplementary Fig. 12). Taken altogether, expression of the β-globin chain by cancer cells is correlated with increased metastatic propensity in mouse models, and with an adverse clinical prognosis in patients.

Recently the role of ROS in blood-borne metastasis was highlighted by a study of patient-derived melanoma xenografts, demonstrating that oxidative stress is a significant hurdle for melanoma dissemination[24]. Two NADPH-generating enzymes, *ALDH1L2* and *MTHFD1*, were identified as suppressors of ROS-mediated toxicity in metastasizing melanoma cells. To extend our studies of *HBB* and obtain a broad overview of changes within the antioxidant machinery in patient-derived primary CTCs, we performed hypergeometric analysis on our single-cell RNA-Seq data from prostate CTCs (Supplementary Fig. 13). While *HBB* is by far the most abundant transcript differentially expressed between CTCs and primary tumours, we identified 16 other antioxidant genes as being significantly

upregulated, including the *ALDH* family members *ALDH9A1* and *ALDH18A1*, as well as *SOD1* and *PRDX* family members (Fig. 5c; Supplementary Table 2). Thus, in addition to *HBB*, multiple antioxidant mechanisms are activated within cancer cells undergoing blood-borne metastasis, supporting the concept that ROS provides a major challenge to the distant spread of cancer.

## Discussion

We have used single-cell RNA-Seq to uncover the unexpected yet frequent expression of β-globin in cancer cells circulating in the bloodstream. The use of human tumour xenografts in a murine background ensures that it is the human tumour-derived transcript that is identified, as opposed to contaminating mouse RBC-derived transcripts. This is also supported by RNA-*in-situ* hybridization to visualize the cellular origin of the *HBB* transcript. Furthermore, the companion α-globin transcript, which is expressed at comparable levels to β-globin in RBCs, is not significantly expressed in CTCs. We demonstrate that *HBB* is specifically induced in cultured epithelial cancer cells by ROS, either following direct hydrogen peroxide treatment or loss of adherence to matrix, which is also known to trigger

increased ROS (refs 3,5). The induction of *HBB* by ROS is mediated at least in part through induction of the newly identified *HBB* regulator *KLF4*, and it is suppressed by the anti-oxidant NAC, consistent with a direct response to oxidative stress. The consequences of ectopic *HBB* expression in cancer cells include reduced oxidative stress, and increased intracellular iron content. Conversely, depletion of *HBB* in cancer cells enhances their sensitivity to ROS and abrogates their ability to form anchorage-independent colonies, a marker of tumour progenitor capacity. Finally, we provide evidence that *HBB* expression by CTCs contributes to their ability to survive in the bloodstream and initiate distant metastases in mouse models. NAC treatment rescues the consequences of *HBB* knockdown on cancer cell survival and metastasis. Although levels of *HBB* expression in primary tumour specimens are much lower than in CTCs, detection of *HBB* mRNA in human tumour specimens is correlated with adverse clinical outcomes.

Studies of cancer metastasis have focused on pathways such as Epithelial-to-Mesenchymal Transition (EMT) that enhance cellular migration, as well as genes that contribute to invasive phenotypes[1]. Such studies have relied primarily on the characterization of genes and pathways whose expression within primary tumours leads to enhanced metastasis. In contrast, the direct analysis of human CTCs reveals additional candidates that appear to be transiently induced during vascular invasion. We have previously documented activation of the non-canonical Wnt pathway and induction of extracellular matrix (ECM)-associated transcripts in CTCs from pancreatic cancer[14,16]. The finding that *HBB* is one of the most significantly upregulated transcript in CTCs across three different cancer types sheds new light on an unexpected pathway contributing to cancer metastasis.

Haemoglobin is generally thought to be exclusively present in cells of the erythroid lineage, with the expression of its α and β chains tightly coordinated. The physiological α2β2 haemoglobin complex has optimal properties for binding oxygen in pulmonary capillaries and releasing it in distal tissues, whereas the abnormal β4 haemoglobin H tetramer (HbH), present in RBCs from patients with alpha-thelassemia, is unstable and has higher affinity for oxygen[25]. In past studies using RT-PCR and microarray-based assays, expression of globin chains has been reported in a number of non-erythroid tissues, including neuronal cells, mesangial cells, macrophages and hepatocytes, as well as in several types of epithelial cancers, including thyroid, cervical and breast tumours[20,26,27]. Although, the functional significance of globin chain expression in these previous reports was uncertain, it was postulated to provide oxygen storage in neurons and to reduce oxidative stress in mesangial cells[28,29]. By applying RNA-Seq technology to human and xenograft-derived CTCs, we were able to show the discoordinated expression pattern of α- and β-globin in these cancer cells, definitively establish the epithelial cancer cell origin of the β-globin transcripts, and begin to explore both the regulation and functional consequences of this aberrant expression.

Whereas, the coregulation of α and β globin gene during RBC development is well established, there are multiple transcription factor binding motifs in the β-globin gene promoter that are absent from that of α-globin[19]. We have shown that the Kruppel-like transcription factor *KLF4*, which binds to the CACCC motif within the β-globin gene promoter, is induced by ROS treatment of epithelial cancer cells, recruited to the *HBB* promoter following ROS, and expressed at high levels in CTCs that also express *HBB* mRNA. This may explain in part the selective induction of the β-globin chain by cellular stress within CTCs. *KLF4* is noteworthy as one of the four Yamanaka factors implicated in stem cell reprogramming[30], and it also plays

important roles in regulating development, differentiation and homoeostasis in various tissues including the intestine and the skin, as well as impacting cell survival and therapeutic resistance in cancer[31,32]. *KLF4* mRNA is also induced by exposure to hydrogen peroxide in vascular smooth muscle cells through a hydroxyl radical-dependent pathway[33]. Other known inducers of β-globin but not α-globin, including interferon-γ (ref. 34), were not differentially expressed in *HBB*-expressing CTCs.

Further biochemical studies will be required to fully characterize the physiological properties of β-globin tetramers as ROS scavenger within CTCs. These transient cell populations constitute a critical step in blood-borne metastasis, which is only now becoming susceptible to molecular analysis. Indeed, the exceptional cellular stress faced by epithelial cancer cells as they lose cell matrix-derived survival signals and circulate in the bloodstream may lead to the activation of unique signalling and anti-apoptotic pathways, which may not be evident through analysis of established primary or metastatic tumour deposits. It is noteworthy that the dramatic over-expression of *HBB* evident in CTCs is not present in either primary tumours or in metastases, but rather reflects the transient state of cancer cells in the vasculature (Supplementary Fig. 11a). As such, this finding is dependent on the development of technologies that now enable the efficient isolation of CTCs with intact RNA, and the application of single-cell RNA sequencing strategies. The recent report of melanoma cell dependence on ROS attenuation to enable blood-borne metastasis is consistent with a broadly relevant phenomenon that may be critical for cancer dissemination[24,35]. Although ubiquitously expressed modulators of ROS, such as *ALDH* family members and glutathione components, contribute to this phenomenon[24], the induction of β-globin and its potential role in mediating sequestration of ROS is a striking illustration of cancer cell hijacking of one of the most restricted and specialized cellular expression programs. The characterization of this CTC survival mechanism may provide novel therapeutic opportunities to suppress cancer metastasis.

## Methods

**Patient samples and CTC isolation.** Patients with a diagnosis of prostate or breast cancer provided informed consent to one of two Institutional Review Board approved protocols (metastatic disease, (DF/HCC 05-300) or localized prostate cancer, (DF/HCC 08-207)). Single CTCs and CTC clusters were isolated from 20 ml fresh whole blood following leucocyte depletion using the microfluidic CTC-iChip[4,13]. Briefly, whole blood samples were spiked with biotinylated antibodies against CD45 (R&D Systems, clone 2D1) and CD66 (AbD Serotec, clone 80H3), followed by incubation with Dynabeads MyOne Streptavidin T1 (Invitrogen) to achieve magnetic depletion of white blood cells. CTC samples isolated from each patient with their GEO accession ID are provided in Supplementary Table 3. Patient information about disease status and therapy is disclosed previously[4,13], and additional information of prostate cancer patients with multiple blood draws is provided in Supplementary Table 4. Primary prostate tumours from 9 patients and metastatic tumours from 19 patients were obtained, sectioned and processed for RNA-ISH.

**Animal models and single-cell RNA-Seq.** Animal experiments were in accordance with institutional guidelines at Massachusetts General Hospital, and approved by the animal protocol (IACUC 2010N000006). An orthotopic lung xenograft model was used to identify differential expression of CTCs compared with primary tumour cells. Briefly, $3 \times 10^5$ MGH134 (ref. 18) cells stably expressing luciferase and GFP were introduced into the lung of NSG mice (male, 8 weeks) by percutaneous injection. Mouse blood samples (1–2 ml) were collected 7–8 weeks after injection, and CTCs were isolated using the microfluidic CTC iChip[16]. Primary tumours were digested into single cells by collagenase and hyaluronidase (Stemcell Technologies). GFP-positive single CTCs and primary tumour cells were individually micromanipulated using a 10 μm transfer tip on an Eppendorf TransferMan NK 2 micromanipulator. Complementary DNA (cDNA) was prepared from single cells, amplified and subjected to library construction for transcriptome analysis using the ABI SOLiD platform, following published protocols[36], with slight modifications[16]. Only cells passing quality control qPCR for

GAPDH and beta-actin were subjected to library construction, followed by sequencing on the ABI 5500XL.

A metastatic prostate cancer mouse model was used to identify *Hbb* expression in CTCs and matched primary and metastatic tumour cells. The CE1-4 cell line stably expressing luciferase and GFP is a clonal cell line derived from the murine castration resistant prostate tumour cell line CE1 (ref. 23). In addition, $1 \times 10^6$ CE1-4 cells were orthotopically introduced into prostate and monitored weekly using the Xenogen IVIS Spectrum *in vivo* imaging system. Mouse blood samples were collected 10–11 weeks after injection, and CTCs were isolated using CTC iChip. Primary and metastatic tumours were digested into single cells by collagenase and hyaluronidase (Stemcell Technologies). GFP-positive single CTCs, primary and metastatic tumour cells were individually micromanipulated using a 10 µm transfer tip on an Eppendorf TransferMan NK 2 micromanipulator. Whole transcriptome amplification was performed using SMARTer v3 Ultra Low Input RNA Kit (Clonetech), and cDNA library preparation was performed using Nextera XT Kit (Illumina). Sequencing was performed on the Illumina HiSeq2500 platform.

**In vivo imaging.** To monitor tumour growth and metastases *in vivo*, BRx50 and H727 cells were engineered to stably express luciferase and GFP. Alternatively, $2 \times 10^5$ BRx50 control cells or *HBB* depleted BRx50 cells were injected into different mammary fad pads within the same mouse (NSG, female, 8weeks). Tumour growth was monitored weekly using the Xenogen IVIS Spectrum *in vivo* imaging system (Caliper Life Sciences). Alternatively, $5 \times 10^4$ BRx50 control cells or *HBB* depleted cells were injected intravenously to evaluate their metastatic potential. Lung metastases were monitored weekly, and mice (NSG, female, 8weeks) were killed after 5 weeks. For NAC rescue experiments, mice were pretreated with NAC ($200 \, mg \, kg^{-1} day^{-1}$) for 3 days before tumour cell inoculation, and then treated daily using the same dose. $2.5 \times 10^5$ H727 cells were used for the intravenous injection model (NSG, male, 8weeks), and mice were killed after 9 weeks. Data analysis was performed using IVIS Lumina Living Image 4.2 following the manufacturer's manual. IVIS luminescence signal was normalized to photon ($p \, s^{-1} \, cm^{-2} \, sr^{-1}$) and total photon count in the region of interest was quantified using the ROI function of the software.

**RNA in-situ hybridization.** CTCs isolated using the CTC-iChip were centrifuged onto poly-L-lysine coated glass slides (Sigma Life Sciences, P0425) for 5 min at 150$g$ using Shandon EZ Megafunnels (A78710001). Slides were fixed with 4% PFA for 10 min, washed with PBS for 5 min and then stored in 100% ethanol at $-20 \, ^\circ C$ until staining. ViewRNA ISH Cell Assay Kit (Affymetrix, Santa Clara, CA) was used to stain CTCs (ref. 4). Briefly, cells were permeabilized using solution QC for 5 min, and then RNA was unmasked using Protease QS (1:2,000 dilution) for 10 min at room temperature (RT). Type 1 probes for *HBB* (VA1-13382) and Type 6 probes for *KRT8* (VA6-11560), *KRT18* (VA6-11561), *KRT19* (VA6-10947), *EPCAM* (VA6-13003) were hybridized to target mRNA for 3 h at 40 °C. Sequential hybridization of Pre-Amplifier molecules, Amplifier molecules, and fluorophore conjugated Label Probe oligonucleotides was performed to amplify the signal, followed by 1-min staining with DAPI (Invitrogen, D3571; $5 \, \mu g \, ml^{-1}$). Slides were then scanned using BioView automated fluorescent imaging platform. A cell is determined as a positive event if it has more than three clear fluorescent dots in one fluorescent channel.

RNA *in situ* Hybridization (RNA-ISH) on tumour tissue was performed using the Affymetrix ViewRNA ISH Tissue Assay Kit (2-plex)[4]. Type 1 probe for *HBB* (VA1-13382) was used at 1:50, and Type 6 probes *KRT8* and *KRT18* were (VA6-11560, VA6-11561) pooled each at 1:200. Images were obtained on the Aperio scanscope system. At least three × 40 images of each tissue core were randomly selected and quantified. Cells with at least three blue dots were classified as epithelial tumour cells, and cells with at least three red dots and three blue dots were classified as *HBB* positive tumour cells.

**ROS staining in CTCs.** The intracellular levels of ROS were measured by staining freshly isolated CTCs with cell-permeant 2′,7′-dichlorodihydrofluorescein diacetate (H2DCFDA). The mitochondrial superoxide levels were measured by staining freshly isolated CTCs with MitoSOX Red (Thermo Fisher).

**Cell lines.** Cell lines used in this study are listed in Supplementary Table 5. Standard cancer cell lines were obtained and authenticated from ATCC (by STR profiling) and maintained as recommended. The lung cancer cell line MGH134 was a gift from J. Engelman and cultured in RPMI-1640 supplemented with 10% fetal bovine serum[18]. Cultured breast CTC cell lines were grown in suspension in ultra-low attachment plates (Corning) in tumour sphere medium (RPMI-1640, EGF ($20 \, ng \, ml^{-1}$), bFGF ($20 \, ng \, ml^{-1}$), 1X B27, 1 × antibiotic/ antimycotic (Life Technologies)) under hypoxic (4% O2) conditions[17]. No mycoplasma contamination has been detected.

**Quantitative proteomics.** CTC lines cultured from blood specimens of multiple breast cancer patients (at multiple time points during the treatment) were collected through centrifugation and the cell pellets were washed with PBS. Multiplexed

quantitative mass spectrometry-based proteomics was performed on an Orbitrap Fusion mass spectrometer by using TMT-10 plex reagents and the SPS-MS3 method (Thermo Scientific)[37]. An estimation of the absolute protein concentration difference between *GAPDH* and *HBB* was done by summing the TMT reporter ion intensities for all MS3 spectra assigned to each protein and calculating the log2 ratio of the two estimated absolute concentrations. The calculation of the concentration difference between *GAPDH* and *HBB* in murine erythrocytes was based on the proteomics data on wild type murine erythrocytes[38].

**Bioinformatic analyses.** RNA-Seq reads were acquired from multiple GEO datasets (GSE74639, GSE51827, GSE55807 and GSE67980). Reads per million (RPM) values were determined from sequencing reads[13]. Single CTCs from CRPC patients were compared with 12 unmatched bulk primary tumour specimens. Single CTCs from NSCLC xenografts were compared with matched single primary tumour cells. Cultured breast CTCs were compared with ATCC established breast cell lines. A *t*-test assuming equal variance in the two classes was then performed for each gene on the log10(RPM) values for the RNA-Seq datasets. The resulting *P* values were used to create False Discover Rate (FDR) estimates by the Benjamini–Hochberg method. A gene was considered differentially expressed if its FDR estimate was < 0.25, and its foldchange was > 2. The constant added to the RPM before taking logarithms was the same for all three datasets.

The correlation analysis of *HBB* expression was performed using the data available from four of the top datasets driving the significant difference in *HBB* expression between primary and metastatic samples (chandran2007a, roessler2015, taylor2010.new and varambally2005), as well as a dataset of publicly available expression from prostate CTCs (ref. 4). Within each data set all pairwise Pearson correlations were calculated between *HBB* and every other assayed gene across all primary and metastatic samples. The Benjamini–Hochberg multiple hypothesis was applied across all experiments and correlations with an FDR < 25% were considered to be significant. Genes that have a significant correlation with *HBB* expression across at least three of the five data sets were selected. Each selected gene was annotated as a chromatin remodeler (GO:0006338)[39], a transcription factor (GO:0003700) (refs 39,40) or histone modification protein[41], if applicable. A list of candidate genes was shown in Supplementary Fig. 5a. These genes together with a group of KLF family members were further screened by evaluating their correlation with *HBB* expression in H727 cells. To test the null hypothesis that the expression of the gene is not correlated with the expression of *HBB* across the specimens shown in Supplementary Fig. 6b, *R*'s cor.test function with method = 'pearson' was used to compute two sided *P* values.

For survival analysis, we used every colorectal, prostate, breast and lung cancer primary tumour gene expression dataset of which we are aware that have survival information for at least 50 unique patients with no treatment before the procedure that obtained the tissue profiled (Supplementary Table 1). If a data set had more than one type of survival information available, we chose the type to use based on the following prioritization in order of the highest priority to lowest: overall survival, death from disease-free survival, distant metastasis-free survival and relapse-free survival. For each data set, we defined specimens for which *HBB* expression was in the upper quartile as *HBB*-high and the rest as *HBB*-low. Then, for each data set we computed a logrank *P* value between the *HBB*-high and *HBB*-low specimens. We estimated a hazard ratio between *HBB*-high and *HBB*-low by fitting a Cox proportional hazards model. False discover rate (FDR) estimates were obtained from the logrank *P* values by the Benjamini–Hochberg method.

To explore the expression of *HBB* in the context of tumour metastases we performed a meta-analysis of *HBB* expression comparing unpaired tumour against metastatic samples from publicly available datasets. Approximately, 19 data sets with both primary and metastatic samples were used in this analysis. A full list of data sets included in the analysis is provided in Supplementary Table 6. The meta-analysis was performed using the meta package of R.

For RNA-Seq results and animal experiments, data represent the mean ± s.e.m. For other cell line experiments, data represent the mean of at least three independent experiments ± s.d. *P* values were determined using the two-tailed Student *t*-test (homoscedastic) and Chi-square test by Graph-Pad Prism 6. For samples with > 3 specimens, data are roughly normal-distributed. An estimate of variation within each group of data is done as part of the *t*-test. A difference was considered statistically significant if the *P* value was equal to or < 0.05. For animal and cell line experiments, no randomization, no blinding and no exclusion were done. The sample size of single-cell RNA-Seq used in this study is determined based on the expense and the availability of data collection, and the need to ensure sufficient statistical power. For animal studies, each mouse group has at least 3 mice.

**shRNAs and siRNAs.** shRNA constructs against *HBB* and non-targeting shRNA control were acquired from the molecular profiling laboratory (MPL) at MGH. Transfection of 293T cells was performed using lipofectamine together with lentiviral packaging plasmids. Lentivirus was collected 48 and 72 h later. H727 and BRx50 cells were infected with lentivirus in the presence of $8 \, \mu g \, ml^{-1}$ polybrene and selected in growth medium containing $2 \, \mu g \, ml^{-1}$ puromycin for 3 days. Transient knockdown of *KLF4*, *KLF6* or *ATF5* in H727 cells was performed using siRNAs purchased from Thermo Fisher Scientific. Samples were harvested 3 days after transfection.

**Primers and quantitative real-time PCR.** Total RNA was extracted using RNeasy Mini Kit (Qiagen). 1 µg of RNA was used to generate cDNA using superscript III First Strand synthesis system (Life Technologies). Reactions were amplified and analysed in triplicate using the ABI 7500 Real-Time PCR System. Primers are listed in Supplementary Table 7.

**Flow cytometry.** Cell cycle analysis was performed using cells fixed and permeabilized by cold 70% ethanol and stained with DAPI. Apoptosis analysis was performed using Annexin V-FITC apoptosis detection kit (Sigma Aldrich). The intracellular levels of ROS were measured by staining live cells with cell-permeant 2′,7′-dichlorodihydrofluorescein diacetate (H2DCFDA). In the indicated experiments, cells were pretreated with 1 mM NAC to rescue ROS induced apoptosis. All flow cytometry experiments were performed using MACSQuant analyser (Miltenyi Biotec) and analysed by FlowJo.

**Cell proliferation and survival assay.** Proliferation was measured using CellTiter-Glo (Promega). Briefly, 4,000 cells were plated in each well in a 96-well plate in a volume of 100 µl growth medium at day 0. At indicated time points, 100 µl of CellTiter-Glo reagent was added into each well and measurements were done using a luminescence plate reader. Similarly, cell survival was measured using the same assay after a 24-h incubation of different doses of hydrogen peroxide. Growth inhibition was determined by the following formula: GI% = 100 (1 − Growth$_{treatment}$/Growth$_{vehicle}$). Soft agar assay was performed to evaluate the long-term growth of cancer cells in suspension. Approximately 50,000 cells were plated in growth medium containing 1.2% methylcellulose in a single well of a 6-well plate. Colonies were stained with 0.005% crystal violet solution after 3 weeks.

**Cell migration and invasion assay.** $2 \times 10^5$ H727 control or *HBB* knockdown cells were seeded in 1%FBS containing medium in the top compartments of Boyden chamber for migration or invasion assay (BD Biosciences). 20 ng ml$^{-1}$ HGF was used as a chemoattractant. After 18 h, cells that migrated or invaded through the membrane were stained with crystal violet and counted using ImageJ.

**ChIP.** DNA and associated proteins on chromatin in cultured cells were crosslinked by 1% formaldehyde for 15 min at 37 °C. Cells were then scraped and collected in cellular lysis buffer (5 mM Pipes, 85 mM KCl, 0.5% NP-40, protease inhibitors). Cytoplasmic lysates were discarded and nuclear components were resuspended in nuclear lysis buffer (50 mM Tris pH 8, 10 mM EDTA pH 8, 0.2% SDS, protease inhibitors) and sonicated for 10 min (Covaris). Approximately 4 µg of KLF4 antibody (Abcam ab151733) or control Rabbit IgG were incubated with 25 µl of protein G magnetic beads for 6 h at 4 °C, and then incubated with 100 µg of cleared chromatin overnight at 4 °C. After three washes, immunoprecipitated material was eluted at 55 °C for 1 h with 10 µg ml$^{-1}$ proteinase K, and then decrosslinked at 65 °C for 4 h (ref. 42). The primer sequences used for ChIP-qPCR are listed in Supplementary Table 7.

**E-cadherin and integrin blocking assay.** Approximately $5 \times 10^5$ H727 cells were trypsinized and treated with 5 µg of antibodies against E-cadherin (MB2) or Integrin B1 (AIIB2-c). RNAs are harvested after different incubation periods (3, 6 and 24 h).

**Iron assay.** Intracellular total Iron levels were measured using Iron Assay Kit from Abcam. Briefly, H727 cells were transiently transfected with pCS2-HBB plasmids for 24 h, and then provided with fresh medium containing 10 µM ($Fe^{2+}$) for 8 h. Cell lysates were then collected and intracellular total Iron levels were measured using the Iron Assay Kit (Abcam) according to manufacture's protocols.

**Intracellular O$_2$ measurement.** Intracellular O$_2$ levels were measured using MITO-ID Intracellular O$_2$ Sensor Probe (Enzo Life Sciences). Briefly, cells were incubated with O$_2$ sensor probes for 16–20 h at 37 °C, and washed with warm standard culture medium right before analysis. Measurements were done using a time resolved plate reader over a period of 60 min.

**Data availability.** Previously deposited RNA-Seq data were assessed from multiple GEO datasets (GSE51827, GSE55807 and GSE67980). Newly generated RNA-Seq data from lung cancer xenografts have been deposited in GEO under accession number GSE74639. RNA-Seq data from prostate cancer allografts are available from the corresponding authors upon request. Other publicly available expression datasets analysed in this study are listed in Supplementary Table 1 and 6. Proteomic raw data have been deposited in the MassIVE proteomics data repository under the accession number MSV000079419. All other remaining data are available within the Article and Supplementary Files, or available form the authors on request.

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

## Acknowledgements

We thank L. Libby, D.B. Fox and O.C. MacKenzie for technical assistance. This work was supported by grants from the Howard Hughes Medical Institute (D.A.H.), National Institute of Biomedical Imaging and Bioengineering (NIBIB) EB008047 (M.T. and D.A.H.), National Cancer Institute (NCI) 2RO1CA129933 (D.A.H.), National Foundation for Cancer Research (D.A.H.), Department of Defense (D.T.M., R.J.L. and D.T.T.), Affymetrix, Inc. (D.T.T., N.D.), Burroughs Wellcome Fund (D.T.T.).

## Author contributions

Y.Z., D.T.M., S.M. and D.A.H. conceived, designed and conducted the study, analysed data, and prepared the manuscript. J.P.S., N.A., N.V.J., M.Y., N.M.K. and V.C. were involved in the conceptual design. B.S.W., R.M. and S.R. performed bioinformatic and statistical analysis. M.T. and N.M.K. provided CTC isolation technology. R.J.L., C.-L.W. and L.V.S. recruited patients and provided clinical samples. J.P.S. developed NSCLC xenograft models. N.V.J. and W.H. performed quantitative proteomics. N.D. and D.T.T. performed RNA-*in situ* hybridization experiments. M.Y. and N.A. helped with *in vivo* experiments. Y.Z., D.T.M., R.D., E.E. and J.D.M. performed most of the other experiments.

## Additional information

**Competing financial interests:** D.T.T. is a paid consultant for Affymetrix, Inc.; R.J.L. is a paid consultant for Janssen LLC. The Massachusetts General Hospital has filed for patent protection for the CTC-iChip technology. The remaining authors declare no competing financial interests.



