## [Peer Review File · Nature Communications]

Reviewers' comments:

Reviewer #1 (Remarks to the Author):

In this report, the authors identified a new machinery utilized by circulating tumor cells (CTCs) to protect their survival against oxidative stress via induction of β -chain of hemoglobin (HBB). The same group has developed methods to isolate CTCs from cancer patients and identified important regulators of metastasis. In the current study, the authors performed single cell RNA-Seq of the CTCs from patient-derived lung cancer xenograft. By comparing the gene expression profiling of the CTCs isolated from the xenograft model with those CTCs from breast and prostate cancer patients (previous studies), HBB is identified as the only gene that is up-regulated in the CTCs compared to primary tumors or cell lines among different cancer types. Through functional assays, the authors demonstrated that induction of HBB in CTCs enhances cell survival under ROS and promotes breast cancer metastasis. In addition, induction of KLF4 is shown to promote expression of HBB under ROS. Finally, the authors revealed a positive correlation between HBB expression and cancer progression/metastasis in patients. ROS has been reported to inhibit distant metastasis of melanoma cells. The current study provided evidence from CTCs to support the concept. The study is well done and the overall conclusion appears to be sound. However, its significance is unclear, since the biological phenomenon revealed is not sufficiently novel, and the design of some experiments is confusing. Especially, some specific points need to be clarified:

1. Design of the experiment that identified HBB as the only up-regulated gene in CTCs among different cancer types is confusing. There are 610 up-regulated genes shared by CTCs from breast and prostate cancer patients. By over-lapping with lung cancer xenograft, HBB is the only up-regulated gene. It is unclear why the authors compared de-regulated genes in lung cancer xenograft, instead of CTCs from lung cancer patients. The authors should compare gene expression profile of CTCs from lung cancer xenograft with those from lung cancer patients, especially HBB expression. In addition, the lung cancer xenograft used is questionable, as it is derived from a TKI-resistant cell line MGH134 with EGFR T790M, which may not represent general features of NSCLCs. More patient-derived xenografts are needed to confirm the key results.
2. It is confusing why majority of the functional assays were performed in a lung carcinoid cancer cell line H727, as the initial gene expression profiling was conducted in CTCs from MGH134-derived xenograft. Gene expression profiles including HBB expression in CTCs from H727-derived xenograft are unknown. Key biological assays in Figure 2 and Figure 3 need to be confirmed in MGH134 and its derived CTCs.
3. The authors claimed that HBB is the only up-regulated gene in CTCs among different cancer types. But they showed later that expression of KLF4 is induced by H₂O₂ in lung cancer and there is enhanced expression of KLF4 in CTCs from breast and prostate cancer (Figure S5E-F). This is unclear and needed to be clarified.
4. Figure S3A, CTCs and ATCC cell lines need to be cultured in the same condition (anchorage independent) when comparing HBB expression among these cells, as the culture condition affects its expression.

Reviewer #2 (Remarks to the Author):

The manuscript describes the molecular analysis of circulating tumor cells (CTCs) using RNA-Seq of CTCs (or clusters thereof) and available metadata to derive hemoglobin β (HBB) as the only gene consistently overexpressed in CTCs from various tumor backgrounds. Using a variety of functional assays in CTC-derived cultures and xenograft models, the effects of depletion and overexpression of HBB are analyzed. The authors conclude that the overexpression of HBB in CTCs protects these cells from oxidative stress to prolong their survival during circulation and thereby increase the likelihood of metastasis formation.

Major points

The manuscript in its current form is characterized by a lack of focus. The erratic use of tumor tissues, cell lines and from patients with various backgrounds (NSCLC, CRC, prostate cancer, breast cancer) leads to a series of descriptive arguments which are not sufficiently persuading and leave the reader confused. For example, different effects on proliferation are seen when using H727 (lung cancer) and LNCaP (prostate cancer) cells (p.9), but this is not commented. The manuscript would be much more convincing if either in vitro and in vivo data would be provided for all tumor backgrounds or if a concise line of arguments and experimental evidence would be provided using one tumor entity only (potentially with short referrals to findings in other tumor backgrounds).

Despite considerable research efforts in the past decade, the importance of CTCs as precursors of metastases is still unclear. While evidence is accumulating that high CTC levels are associated with aggressive tumors and poor patient prognosis in several cancer entities, the relevance of the finding with respect to the formation of metastases through CTCs is highly debated. In the current study, this is reflected e. g. by the finding that the expression of HBB in CTCs is clearly distinct from that in primary tumors and metastases (Fig. 1A: heterogeneous expression of HBB in CTCs, esp. in lung xenografts and prostate cancer; fig. S9A). How is the metastasis formation process on a molecular level explained in the light of these data?

The inoculation tests of BRx50 cells into mouse models appear to be quite convincing. However, the pre-treatment of those animals with antioxidants might cause a multitude of different effects which have nothing to do with the CTC models used.

The functional analysis of the relationship between oxidative stress and HBB is done by using H727 cell lines as models (Fig. 3). However, to support the conclusions, this experiment should not be performed in an epithelial cancer cell line but in the CTCs. The evidence for ROS protective effects of HBB in CTCs is not formally provided (as the authors themselves concede (Discussion, lines 373, 374), and the analysis of genes involved in oxygen detoxification is not sufficient. It also remains unexplained how the cytoprotective effect against ROS is conferred in the absence of HBA protein.

Technical issues:

How homogeneous the BRx50 cells? Are they characteristic for breast cancer CTCs in general? How do they behave upon cell culturing?

How heterogeneous is the cell population after the application of the CTC-iChip?

How high is the rate of contaminating non tumor-epithelial cells?

Minor points:

Fig. 4: The title is misleading ("breast CTCs"): The figure contains data from breast, multiple (D), and prostate cancers (E,F).

Fig. 4F (prostate cancer) For many of the antioxidant genes, there appears to indicate a bi-modal expression CTCs. How do the authors explain this finding?

Reviewer #3 (Remarks to the Author):

The beta-chain of hemoglobin (HBB) is a component of the oxygen transport machinery of red blood cells. In this manuscript the authors compared single cell RNA-Seq profiles of circulating tumor cells (CTC) from different cancers and found induction of expression of HBB as a common feature. They show that HBB increases intracellular iron levels and suppresses ROS. Depletion of HBB in CTC-derived cultures sensitized them to ROS-mediated cell death, abrogated colony formation and reduced metastases.

While the RNA-Seq data leading to identification of HBB as a component/maker of CTC and the in

vivo data are convincing, this manuscript lacks mechanistic insight of how HBB expression is regulated. After treatment of cell lines with hydrogen peroxide the authors conclude that ROS is a driving force in HBB expression. However this is unclear since ectopic administration of hydrogen peroxide disturbs membranes and cell-cell or cell-matrix contacts and effects observed may not be due to intrinsic increase oxidative stress.

The authors should demonstrate the increase of ROS in single and clustered CTCs, i.e. as shown in Figure 1B for HBB. Another weakness is that the source of intracellular ROS is not defined for CTCs.

In Fig. 2, adherent cell lines have been investigated for drivers of HBB expression. However, data with H727 indicate that treatment with H₂O₂ drives HBB expression. Since serum deprivation which generates ROS had no effect and suspension only a marginal effect it may be concluded that ROS-induced detachment or interruption of cell-cell or cell-matrix contacts are driving forces. This should be further tested with use of integrin blocking antibodies and E-cadherin blocking antibodies in a time course experiment.

How does hydrogen peroxide activate KLF?

Reviewer #4 (Remarks to the Author):

Recommending reject outright on the grounds that there is insufficient novelty and, although there is considerable work involved, the manuscript does not provide sufficient conceptual advance to warrant publication in Nature Communications.

The author's describe the detection of high levels of the β -chain of hemoglobin (HBB) RNA in circulating tumor cells (CTCs) from breast, prostate and lung cancers. They go on to show that HBB is induced by exposure to reactive oxygen species (ROS) or by loss of adhesion and that the transcriptional regulator KLF4 is involved in the induction. To follow up on mechanism they show that reduction of HBB can increase apoptosis following ROS exposure and that ectopic expression of HBB can ameliorate the ROS induced apoptosis. These observations were followed up by mouse in vivo studies and analysis of public clinical databases which support a link between HBB and metastatic potential as well as high levels of HBB in the tumor being a significant predictor for a poor clinical outcome.

Although the authors present all of these findings as being novel most of the key elements such as HBB being induced by exposure to ROS, ectopic HBB reducing ROS induced apoptosis and the link with metastasis spread were clearly described in previous publications referenced by the authors (particularly refs 20 and 21 abstracts shown below). It is not clear if the authors are unaware of the content of the papers they reference but statements such "The unexpected expression of globin genes by some epithelial cancers has been described (20-22), although the functional consequences of globin chain expression have not been clearly demonstrated." on page 9 are clearly misleading. In a similar example on Page 9 the authors state "Our observation that hydrogen peroxide specifically induces HBB expression suggests a role for β -globin in ROS quenching, a notion that has been suggested for mesangial cells of the normal kidney (23)." but do not reference ref 21 which in the abstract states: "Hgb is expressed in cervical carcinoma cells and may act as an antioxidant, attenuating oxidative stress-induced damage in cervical cancer cells."

Authors reference 20

Capulli, M., A. Angelucci, K. Driouch, T. Garcia, P. Clement-Lacroix, F. Martella, N. Rucci (2012). "Increased expression of a set of genes enriched in oxygen binding function discloses a predisposition of breast cancer bone metastases to generate metastasis spread in multiple organs." *J Bone Miner Res* 27(11): 2387-2398.

Bone is the preferential site of distant metastasis in breast carcinoma (BrCa). Patients with metastasis restricted to bone (BO) usually show a longer overall survival compared to patients

who rapidly develop multiple metastases also involving liver and lung. Hence, molecular predisposition to generate bone and visceral metastases (BV) represents a clear indication of poor clinical outcome. We performed microarray analysis with two different chip platforms, Affymetrix and Agilent, on bone metastasis samples from BO and BV patients. The unsupervised hierarchical clustering of the resulting transcriptomes correlated with the clinical progression, segregating the BO from the BV profiles. Matching the twofold significantly regulated genes from Affymetrix and Agilent chips resulted in a 15-gene signature with 13 upregulated and two downregulated genes in BV versus BO bone metastasis samples. In order to validate the resulting signature, we isolated different MDA-MB-231 clonal subpopulations that metastasize only in the bone (MDA-BO) or in bone and visceral tissues (MDA-BV). Six of the signature genes were also significantly upregulated in MDA-BV compared to MDA-BO clones. A group of upregulated genes, including Hemoglobin B (HBB), were involved in oxygen metabolism, and in vitro functional analysis of HBB revealed that its expression in the MDA subpopulations was associated with a reduced production of hydrogen peroxide. Expression of HBB was detected in primary BrCa tissue but not in normal breast epithelial cells. Metastatic lymph nodes were frequently more positive for HBB compared to the corresponding primary tumors, whereas BO metastases had a lower expression than BV metastases, suggesting a positive correlation between HBB and ability of bone metastasis to rapidly spread to other organs. We propose that HBB, along with other genes involved in oxygen metabolism, confers a more aggressive metastatic phenotype in BrCa cells disseminated to bone.

Authors reference 21

Li, X., Z. Wu, Y. Wang, Q. Mei, X. Fu and W. Han (2013). "Characterization of adult alpha- and beta-globin elevated by hydrogen peroxide in cervical cancer cells that play a cytoprotective role against oxidative insults." PLoS One 8(1): e54342.

OBJECTIVES: Hemoglobin (Hgb) is the main oxygen and carbon dioxide carrier in cells of erythroid lineage and is responsible for oxygen delivery to the respiring tissues of the body. However, Hgb is also expressed in nonerythroid cells. In the present study, the expression of Hgb in human uterine cervix carcinoma cells and its role in cervical cancer were investigated. **METHODOLOGY:** The expression level of Hgb in cervical cancer tissues was assessed by quantitative reverse transcriptase-PCR (qRT-PCR). We applied multiple methods, such as RT-PCR, immunoblotting, and immunohistochemical analysis, to confirm Hgb expression in cervical cancer cells. The effects of ectopic expression of Hgb and Hgb mutants on oxidative stress and cell viability were investigated by cellular reactive oxygen species (ROS) analysis and lactate dehydrogenase (LDH) array, respectively. Both Annexin V staining assay by flow cytometry and caspase-3 activity assay were used, respectively, to evaluate cell apoptosis. **RESULTS:** qRT-PCR analysis showed that Hgb-alpha (HBA1) and Hgb-beta-globin (HBB) gene expression was significantly higher in cervical carcinoma than in normal cervical tissues, whereas the expression of hematopoietic transcription factors and erythrocyte specific marker genes was not increased. Immunostaining experiments confirmed the expression of Hgb in cancer cells of the uterine cervix. Hgb mRNA and protein were also detected in the human cervical carcinoma cell lines SiHa and CaSki, and Hgb expression was up-regulated by hydrogen peroxide-induced oxidative stress. Importantly, ectopic expression of wild type HBA1/HBB or HBA1, rather than mutants HBA1(H88R)/HBB(H93R) unable to bind hemo, suppressed oxidative stress and improved cell viability. **CONCLUSIONS:** The present findings show for the first time that Hgb is expressed in cervical carcinoma cells and may act as an antioxidant, attenuating oxidative stress-induced damage in cervical cancer cells. These data provide a significant impact not only in globin biology but also in understanding of cervical cancer pathogenesis associated with oxidative stress.

Additional comments:

On Page 8, it is stated "Out of 21 candidates, only three genes, KLF4, KLF6 and ATF5, are induced when H727 cells are treated with hydrogen peroxide or grown under anchorage-free conditions (Figures S5B, C)." yet the figure S5B shows 6 genes are statistically induced by hydrogen peroxide. Please clarify.

On Page 6 "The specific overexpression of HBB, but not HBA, is also evident using single cell RNA-sequencing of CTCs and CTC-clusters from multiple independent breast and prostate cancer patients (Figures 1A,D,E)." However, only 1A includes HBA data.

On Page 12, "HBB mRNA level in primary tumor samples is generally low (0-0.1%), while increased expression is detected in 7 out of 19 metastatic prostate tumors (6.3-18.1%) (Figures 4E, S8)" However, the Figures referred to do not support this statement since 4E = localised 0.9%, met 15.9% and S8 shows RNA in situ.

On Page 12 "Expression of HBB is low in primary tumor cells, markedly induced in CTCs, but is sustained at an intermediate level in metastases (Figure S9A, B)." Is there a statistical difference between PT and Met in S9A?

Reviewer #1 (Remarks to the Author):

In this report, the authors identified a new machinery utilized by circulating tumor cells (CTCs) to protect their survival against oxidative stress via induction of β -chain of hemoglobin (HBB). The same group has developed methods to isolate CTCs from cancer patients and identified important regulators of metastasis. In the current study, the authors performed single cell RNA-Seq of the CTCs from patient-derived lung cancer xenograft. By comparing the gene expression profiling of the CTCs isolated from the xenograft model with those CTCs from breast and prostate cancer patients (previous studies), HBB is identified as the only gene that is up-regulated in the CTCs compared to primary tumors or cell lines among different cancer types. Through functional assays, the authors demonstrated that induction of HBB in CTCs enhances cell survival under ROS and promotes breast cancer metastasis. In addition, induction of KLF4 is shown to promote expression of HBB under ROS. Finally, the authors revealed a positive correlation between HBB expression and cancer progression/metastasis in patients. ROS has been reported to inhibit distant metastasis of melanoma cells. The current study provided evidence from CTCs to support the concept. The study is well done and the overall conclusion appears to be sound. However, its significance is unclear, since the biological phenomenon revealed is not sufficiently novel, and the design of some experiments is confusing. Especially, some specific points need to be clarified:

1. Design of the experiment that identified HBB as the only up-regulated gene in CTCs among different cancer types is confusing. There are 610 up-regulated genes shared by CTCs from breast and prostate cancer patients. By over-lapping with lung cancer xenograft, HBB is the only up-regulated gene. It is unclear why the authors compared de-regulated genes in lung cancer xenograft, instead of CTCs from lung cancer patients. The authors should compare gene expression profile of CTCs from lung cancer xenograft with those from lung cancer patients, especially HBB expression. In addition, the lung cancer xenograft used is questionable, as it is derived from a TKI-resistant cell line MGH134 with EGFR T790M, which may not represent general features of NSCLCs. More patient-derived xenografts are needed to confirm the key results.

We agree with the reviewer that the emergence of HBB as a consistently upregulated CTC transcript emerged from multiple comparisons in different cancer types. For technical reasons, we have not been able to isolate sufficient single CTCs from patients with lung cancer (a challenge in the field most likely due to low cell surface markers in lung CTCs) and hence we used a human GFP-tagged lung cancer PDX mouse model, comparing the primary tumor cells to the CTCs. This proved to be particularly helpful in subsequently demonstrating that the HBB transcript is tumor derived (ie human) rather than from mouse RBC contamination. In response to the reviewer comments, we have revised the text to clarify that HBB is not the only upregulated gene in CTCs from all these cancer types, but rather that it stands out because it is so unexpected and so consistently upregulated. We agree with the reviewer that this represents the data more clearly and explains our decision to focus this study on the significance of HBB expression in CTCs. We have revised the texts accordingly.

2. It is confusing why majority of the functional assays were performed in a lung carcinoid cancer cell line H727, as the initial gene expression profiling was conducted in CTCs from MGH134-derived xenograft. Gene expression profiles including HBB expression in CTCs from H727-derived xenograft are unknown. Key biological assays in Figure 2 and Figure 3 need to be confirmed in MGH134 and its derived CTCs.

We had pursued multiple cancer cell lines to demonstrate the broad applicability of our findings, but we certainly appreciate the reviewer's concern that in the previous manuscript, we had used different cell lines in the various figures. We have generated new data to confirm the key results in Figure 2 and 3, using the breast cancer patient-derived CTC culture BRx50, and we now present these cells as the primary source of data in the figures (other cell lines are now in supplementary). We show that the up-regulation of HBB by exposure to H₂O₂ is blocked by pretreating BRx50 cells with NAC, and that it is primarily mediated by KLF4 (Figures 2A-C). We also demonstrate that depletion of HBB in BRx50 cells significantly increases apoptosis following ROS exposure, and abrogates anchorage-independent colony formation, while ectopic overexpression of HBB increases intracellular iron (Fe) levels and suppresses ROS (Figure 3). The relevant new data are shown below. We still highlight the H727 cells for anchorage independence experiments, since the CTC cultures grow in suspension at baseline, whereas the H727 cells are adherent and show induction of both ROS and HBB expression upon loss of matrix adherence. We also have revised the text extensively to clarify the use of these different cell lines.

Figure 2

Figure 3

3. The authors claimed that HBB is the only up-regulated gene in CTCs among different cancer types. But they showed later that expression of KLF4 is induced by H₂O₂ in lung

cancer and there is enhanced expression of KLF4 in CTCs from breast and prostate cancer (Figure S5E-F). This is unclear and needed to be clarified.

As noted above, many genes are upregulated in CTCs, and we focused on HBB because of the unexpected nature of this transcript and the fact that it is overexpressed so frequently and to such high levels. We have now clarified this in the text. The HBB regulator KLF4 is significantly elevated in CTCs from prostate cancer patients, and it is one of the up-regulated genes in CTCs in both breast and prostate datasets. However, the up-regulation of KLF4 in CTCs from NSCLC xenografts is not statistically significant, and it is probably due to the limited sample size of single cells isolated from NSCLC xenografts (10 lung CTCs and 6 primary tumor cells), where only 52 genes passed the statistical threshold. We have revised the text to clarify (as in response to point #1) that we selected HBB for further study because of its biological interest and high frequency of upregulation, rather than it being the sole statistically significant transcript in all tumor types.

4. Figure S3A, CTCs and ATCC cell lines need to be cultured in the same condition (anchorage independent) when comparing HBB expression among these cells, as the culture condition affects its expression.

We now show data more clearly for 3 different cell lines (CTCs which grow in suspension and two lung cancer cell lines that grow attached to plastic). The conditions of culture are intrinsic to the cells studied. Cultured CTCs only survive and proliferate in suspension culture, and under these conditions, they express elevated levels of HBB. Two lung cancer cell lines tested grow as attached 2D cultures which have low-level HBB expression under these conditions, show induction of HBB upon loss of adhesion. This is one of the most compelling observations pointing to the fact that HBB expression is not a contaminant from RBCs or a cell type-specific anomaly. Rather it is inducible within a given cell type and is associated with ROS stress or loss of matrix adhesion (which is also a known source of ROS stress). We have revised the text to clarify these points.

Reviewer #2 (Remarks to the Author):

The manuscript describes the molecular analysis of circulating tumor cells (CTCs) using RNA-Seq of CTCs (or clusters thereof) and available metadata to derive hemoglobin b (HBB) as the only gene consistently overexpressed in CTCs from various tumor backgrounds. Using a variety of functional assays in CTC-derived cultures and xenograft models, the effects of depletion and overexpression of HBB are analyzed. The authors conclude that the overexpression of HBB in CTCs protects these cells from oxidative stress to prolong their survival during circulation and thereby increase the likelihood of metastasis formation.

Major points

The manuscript in its current form is characterized by a lack of focus. The erratic use of tumor tissues, cell lines and from patients with various backgrounds (NSCLC, CRC, prostate cancer, breast cancer) leads to a series of descriptive arguments which are not sufficiently persuading and leave the reader confused. For example, different effects on proliferation are seen when using H727 (lung cancer) and LNCaP (prostate cancer) cells

(p.9), but this is not commented. The manuscript would be much more convincing if either in vitro and in vivo data would be provided for all tumor backgrounds or if a concise line of arguments and experimental evidence would be provided using one tumor entity only (potentially with short referrals to findings in other tumor backgrounds).

We agree with the reviewer and apologize for the confusing presentation of different cell lines in the originally submitted manuscript. As noted in response to reviewer 1 (point #2), we now focus primarily on the cultured CTCs (BRx50) and have repeated all experiments with that cell line, while showing other cell lines in supplementary data. We also show the H727 cell data where appropriate to demonstrate the potent effect of loss of matrix adherence (H727 grow on 2D plastic and hence can be shown to acquire HBB expression upon growth in suspension, whereas CTCs already grow in suspension and have baseline HBB expression). In addition to extensively reorganizing the manuscript, we have added new data demonstrating ROS levels in primary CTCs (Figures 1F-G), NAC inhibition of effects caused by HBB depletion in CTC cultures (Figure 3), and the regulation of HBB in part through KLF4 (Figure 2).

Despite considerable research efforts in the past decade, the importance of CTCs as precursors of metastases is still unclear. While evidence is accumulating that high CTC levels are associated with aggressive tumors and poor patient prognosis in several cancer entities, the relevance of the finding with respect to the formation of metastases through CTCs is highly debated. In the current study, this is reflected e.g. by the finding that the expression of HBB in CTCs is clearly distinct from that in primary tumors and metastases (Fig. 1A: heterogeneous expression of HBB in CTCs, esp. in lung xenografts and prostate cancer; fig. S9A). How is the metastasis formation process on a molecular level explained in the light of these data?

The role of tumor cells in the blood circulation as mediators of blood-borne metastasis is of course evident, but we agree with the reviewer that the vast majority of CTCs are likely to die in the bloodstream and never give rise to blood borne metastases. That is a central issue for the field, as the reviewer points out. It also underscores the heterogeneity among CTCs (which is why we focus on single CTC RNA sequencing) and also the impact and significance of stress (including oxidative stress, loss of matrix adhesion) which likely contributes to the death of cancer cells in the bloodstream. It is in this context that our observations highlight the ROS stress placed on cancer cells in the blood and a novel and unexpected mechanism that helps circumvent this stress. We have revised the manuscript to make this point more clear and place it in context.

The inoculation tests of BRx50 cells into mouse models appear to be quite convincing. However, the pre-treatment of those animals with antioxidants might cause a multitude of different effects which have nothing to do with the CTC models used.

We agree with the reviewer that anti-oxidant treatment of mice could cause a number of effects, in addition to suppressing blood-based ROS. In this context, however, we note that the NAC effect is also evident in vitro where it rescues the effect of HBB knockdown or hydrogen peroxide treatment. We also note while our work was in preparation, a recent publication from Sean Morrison's lab pointing similarly to NAC treatment enhancing the metastatic (but not primary tumor) formation by primary melanoma cells (PMID: 26466563). In response to the reviewer, we now acknowledge that NAC treatment of mice could have additional effects.

The functional analysis of the relationship between oxidative stress and HBB is done by using H727 cell lines as models (Fig. 3). However, to support the conclusions, this experiment should not be performed in an epithelial cancer cell line but in the CTCs. The evidence for ROS protective effects of HBB in CTCs is not formally provided (as the authors themselves concede (Discussion, lines 373, 374), and the analysis of genes involved in oxygen detoxification is not sufficient. It also remains unexplained how the cytoprotective effect against ROS is conferred in the absence of HBA protein.

We appreciate the reviewer's suggestion and have now performed all these experiments using the breast CTC cultures (BRx50), with other tumor cell lines shown in supplementary (see responses above). We now demonstrate that depletion of HBB in cultured CTCs significantly suppresses long-term colony formation in soft agar, short-term proliferation, accompanied by increases apoptosis and intracellular ROS level (Figures 3A-E). On the other hand, ectopic overexpression of HBB in cultured CTCs suppresses ROS and increases intracellular iron (Fe) levels (Figures 3G-I). We also discuss a potential mechanism whereby sequestration of iron (Fe) by HBB homotetramers would lessen the availability of iron in radical-generating reactions.

Figure 3

Technical issues:

How homogeneous the BRx50 cells? Are they characteristic for breast cancer CTCs in general? How do they behave upon cell culturing?

How heterogeneous is the cell population after the application of the CTC-iChip?

How high is the rate of contaminating non tumor-epithelial cells?

BRx50 cells were cultured from the blood sample of a patient with metastatic estrogen receptor (ER)-positive breast cancer. This cell line expresses abundant levels of Keratin 8, Keratin 18, EpCAM and estrogen receptor (ESR1), confirming their luminal origin. Mutation analysis shows a mutation in ESR1 (L536P) but no mutation in PI3KCA. Patient derived CTCs only grow under suspension in the medium containing B27, EGF, FGF, without FBS (PMID: 25013076). While we have recently demonstrated

heterogeneous tumor subpopulations within CTC cultures (PMID: 27556950), these are long term suspension cultures that are free of any contaminating hematopoietic or fibroblast cells (as demonstrated by both protein staining and single cell sequencing). In contrast to these purified CTC cultures, primary CTCs isolated from the CTC-iChip do have contaminating hematopoietic cells (approximately 500 WBCs in output per ml of whole blood processed, reflecting a 10^{4-5} purification of CTCs from whole blood). For these reasons all data from primary (non-cultured) CTCs is derived from single cells that are viably stained for cell surface epithelial marker expression, micromanipulated for selection, and subjected to single cell RNA sequencing (which in addition to HBB expression, confirms lineage of each cell). For reference, the rate of non-tumor-epithelial cells identified by the CTC-iChip device is extremely low in healthy donors (mean \pm SD, $0.17 \pm 0.12/\text{ml}$), while the range of CTCs detected from 37 prostate cancer patients is 0.5 to 610/ml with a mean of 50.3/ml (PMID: 23552373). Expression of lineage markers (eg. PSA RNA in prostate CTCs) is used to confirm that the single cell studied is indeed a tumor-derived cell (The prostate has been resected in patients with prostate cancer, and all PSA transcripts are derived from metastatic cancer cells.).

Minor points:

Fig. 4: The title is misleading ("breast CTCs"): The figure contains data from breast, multiple (D), and prostate cancers (E, F).

We apologize and have revised the figure and title.

Fig. 4F (prostate cancer) For many of the antioxidant genes, there appears to indicate a bi-modal expression CTCs. How do the authors explain this finding?

We agree with the reviewer's observation regarding bi-modal expression pattern of anti-oxidant genes in CTCs. As we reported previously, there is considerably intercellular heterogeneity in CTCs within individual cancer patients and even higher diversity in CTCs across different patients (PMID: 26383955). It would be of considerable interest whether this heterogeneity in anti-oxidant gene expression patterns is correlated with the ability of CTCs to survive under oxidative stress as they circulate in the bloodstream.

Reviewer #3 (Remarks to the Author):

The beta-chain of hemoglobin (HBB) is a component of the oxygen transport machinery of red blood cells. In this manuscript the authors compared single cell RNA-Seq profiles of circulating tumor cells (CTC) from different cancers and found induction of expression of HBB as a common feature. They show that HBB increases intracellular iron levels and suppresses ROS. Depletion of HBB in CTC-derived cultures sensitized them to ROS-mediated cell death, abrogated colony formation and reduced metastases.

While the RNA-Seq data leading to identification of HBB as a component/maker of CTC and the in vivo data are convincing, this manuscript lacks mechanistic insight of how HBB expression is regulated. After treatment of cell lines with hydrogen peroxide the authors conclude that ROS is a driving force in HBB expression. However this is unclear since ectopic administration of hydrogen peroxide disturbs membranes and cell-cell or cell-matrix contacts and effects observed may not be due to intrinsic increase oxidative stress. The authors should demonstrate the increase of ROS in single and clustered CTCs, i.e. as shown in Figure 1B for HBB. Another weakness is that the source of

intracellular ROS is not defined for CTCs.

We thank the reviewer for these suggestions and we have now added new data showing high ROS staining (by DCF staining) within primary single and clustered CTCs directly isolated from blood specimens of prostate cancer patients (Figures 1F-G). Low DCF staining is infrequently observed within the normal WBCs that contaminate the microfluidic CTC-iChip product. In contrast, MitoSOX staining of mitochondria is high in most CTCs and WBCs, suggesting mitochondria are functional in both CTCs and WBCs. We show these data below (revised figure 1) and we have revised the text accordingly.

In Fig. 2, adherent cell lines have been investigated for drivers of HBB expression. However, data with H727 indicate that treatment with H₂O₂ drives HBB expression. Since serum deprivation which generates ROS had no effect and suspension only a marginal effect it may be concluded that ROS-induced detachment or interruption of cell-cell or cell-matrix contacts are driving forces. This should be further tested with use of integrin blocking antibodies and E-cadherin blocking antibodies in a time course experiment.

Again, we thank the reviewer for this suggestion, and we now add new data reflecting these requested experiments. We used blocking antibodies against E-cadherin or Integrin B1 to disrupt cell-cell adhesion in H727 cells, but did not observe an increase in HBB RNA. This is now shown in Figure S4C (and pasted below). We therefore suggest that it is the increased ROS associated with anchorage independent growth, rather than loss of cell-cell contact itself, which drives the induction of HBB RNA. This is also consistent with the effect of NAC in blocking the upregulation of HBB levels in H727 cells caused by growth in suspension. Serum deprivation may also induce some ROS, but typically this occurs after more prolonged exposure than the 3 hour treatment time we used here.

How does hydrogen peroxide activate KLF?

Upregulation of KLF4 mRNA by hydrogen peroxide has also been observed in vascular smooth muscle cells, where it is mediated through hydroxyl radicals, p38MAP kinase-, calcium-, and protein synthesis-dependent pathways (PMID: 12087069). We now cite this reference in the text.

Reviewer #4 (Remarks to the Author):

Recommending reject outright on the grounds that there is insufficient novelty and, although there is considerable work involved, the manuscript does not provide sufficient conceptual advance to warrant publication in Nature Communications.

The author's describe the detection of high levels of the β -chain of hemoglobin (HBB) RNA in circulating tumor cells (CTCs) from breast, prostate and lung cancers. They go on to show that HBB is induced by exposure to reactive oxygen species (ROS) or by loss of adhesion and that the transcriptional regulator KLF4 is involved in the induction. To follow up on mechanism they show that reduction of HBB can increase apoptosis following ROS exposure and that ectopic expression of HBB can ameliorate the ROS induced apoptosis. These observations were followed up by mouse in vivo studies and analysis of public clinical databases which support a link between HBB and metastatic potential as well as high levels of HBB in the tumor being a significant predictor for a poor clinical outcome.

Although the authors present all of these findings as being novel most of the key elements such as HBB being induced by exposure to ROS, ectopic HBB reducing ROS induced apoptosis and the link with metastasis spread were clearly described in previous publications referenced by the authors (particularly refs 20 and 21 abstracts shown below). It is not clear if the authors are unaware of the content of the papers they reference but statements such "The unexpected expression of globin genes by some epithelial cancers has been described (20-22), although the functional consequences of globin chain expression have not been clearly demonstrated." on page 9 are clearly misleading. In a similar example on Page 9 the authors state "Our observation that hydrogen peroxide specifically induces HBB expression suggests a role for β -globin in ROS quenching, a notion that has been suggested for mesangial cells of the normal kidney (23)." but do not reference ref 21 which in the abstract states: "Hgb is expressed in cervical carcinoma cells and may act as an antioxidant, attenuating oxidative stress-induced damage in cervical cancer cells."

Authors reference 20

Capulli, M., A. Angelucci, K. Driouch, T. Garcia, P. Clement-Lacroix, F. Martella, N. Rucci (2012). "Increased expression of a set of genes enriched in oxygen binding function discloses a predisposition of breast cancer bone metastases to generate metastasis spread in multiple organs." *J Bone Miner Res* 27(11): 2387-2398. Bone is the preferential site of distant metastasis in breast carcinoma (BrCa). Patients with metastasis restricted to bone (BO) usually show a longer overall survival compared to patients who rapidly develop multiple metastases also involving liver and lung. Hence, molecular predisposition to generate bone and visceral metastases (BV) represents a clear indication of poor clinical outcome. We performed microarray analysis with two

different chip platforms, Affymetrix and Agilent, on bone metastasis samples from BO and BV patients. The unsupervised hierarchical clustering of the resulting transcriptomes correlated with the clinical progression, segregating the BO from the BV profiles. Matching the twofold significantly regulated genes from Affymetrix and Agilent chips resulted in a 15-gene signature with 13 upregulated and two downregulated genes in BV versus BO bone metastasis samples. In order to validate the resulting signature, we isolated different MDA-MB-231 clonal subpopulations that metastasize only in the bone (MDA-BO) or in bone and visceral tissues (MDA-BV). Six of the signature genes were also significantly upregulated in MDA-BV compared to MDA-BO clones. A group of upregulated genes, including Hemoglobin B (HBB), were involved in oxygen metabolism, and in vitro functional analysis of HBB revealed that its expression in the MDA subpopulations was associated with a reduced production of hydrogen peroxide. Expression of HBB was detected in primary BrCa tissue but not in normal breast epithelial cells. Metastatic lymph nodes were frequently more positive for HBB compared to the corresponding primary tumors, whereas BO metastases had a lower expression than BV metastases, suggesting a positive correlation between HBB and ability of bone metastasis to rapidly spread to other organs. We propose that HBB, along with other genes involved in oxygen metabolism, confers a more aggressive metastatic phenotype in BrCa cells disseminated to bone.

Authors reference 21

Li, X., Z. Wu, Y. Wang, Q. Mei, X. Fu and W. Han (2013). "Characterization of adult alpha- and beta-globin elevated by hydrogen peroxide in cervical cancer cells that play a cytoprotective role against oxidative insults." PLoS One 8(1): e54342.

OBJECTIVES: Hemoglobin (Hgb) is the main oxygen and carbon dioxide carrier in cells of erythroid lineage and is responsible for oxygen delivery to the respiring tissues of the body. However, Hgb is also expressed in nonerythroid cells. In the present study, the expression of Hgb in human uterine cervix carcinoma cells and its role in cervical cancer were investigated. **METHODOLOGY:** The expression level of Hgb in cervical cancer tissues was assessed by quantitative reverse transcriptase-PCR (qRT-PCR). We applied multiple methods, such as RT-PCR, immunoblotting, and immunohistochemical analysis, to confirm Hgb expression in cervical cancer cells. The effects of ectopic expression of Hgb and Hgb mutants on oxidative stress and cell viability were investigated by cellular reactive oxygen species (ROS) analysis and lactate dehydrogenase (LDH) array, respectively. Both Annexin V staining assay by flow cytometry and caspase-3 activity assay were used, respectively, to evaluate cell apoptosis. **RESULTS:** qRT-PCR analysis showed that Hgb-alpha- (HBA1) and Hgb-beta-globin (HBB) gene expression was significantly higher in cervical carcinoma than in normal cervical tissues, whereas the expression of hematopoietic transcription factors and erythrocyte specific marker genes was not increased. Immunostaining experiments confirmed the expression of Hgb in cancer cells of the uterine cervix. Hgb mRNA and protein were also detected in the human cervical carcinoma cell lines SiHa and CaSki, and Hgb expression was up-regulated by hydrogen peroxide-induced oxidative stress. Importantly, ectopic expression of wild type HBA1/HBB or HBA1, rather than mutants HBA1(H88R)/HBB(H93R) unable to bind hemo, suppressed oxidative stress and improved cell viability. **CONCLUSIONS:** The present findings show for the first time that Hgb is expressed in cervical carcinoma cells and may act as an antioxidant, attenuating oxidative stress-induced damage in cervical cancer cells. These data provide a significant impact not only in globin biology but also in understanding of cervical cancer pathogenesis associated with oxidative stress.

We respectfully disagree with the reviewer regarding the novelty of our findings. It is always difficult to strike the right balance between referencing past literature (much of which may be incomplete or using older technologies) and clearly presenting the novelty of current work. We have endeavored to strike a better balance in the revised manuscript to satisfy the reviewer. We had cited all the relevant literature that we could identify, and it consisted primarily of tissue section analyses and some cancer cell line-based experiments. However, to argue that it is already established in the field that cancer cells upregulate beta-globin as a mechanism of dealing with ROS as they circulate through the bloodstream (ie our observation) is quite unfair and unreasonable. The comments of the other reviewers and the responses that we have received to presentations of this work are consistent with it being a highly unexpected observation, rather than “old news” as suggested by the reviewer. The novelty of our work, compared to past experiments include the following:

- (1) High-level expression of beta globin in CTCs (which dwarfs the low levels observed in primary and metastatic cancers). No past study has examined CTCs and the few past reports described low-level expression of both globin.
- (2) Demonstration that beta and not alpha globin is induced in cancer cells and most importantly, definitive evidence (using human/mouse polymorphism sequencing) that the HBB transcript is derived from tumor cells and not a contaminant from the abundant red blood cells (this has never been shown previously, and presumably the limited impact of the past tumor-based studies in the literature may have resulted from suspicion that the globin transcripts were blood cell derived contaminants)
- (3) Demonstration of the mechanism of HBB induction, from anchorage independent growth to ROS induction (and its reversal by NAC), and its link to KLF4 transcription factor regulation and HBB promoter binding in vivo. (these observations and this level of analysis are totally novel. Previous functional studies were very limited in scope and used high-level ectopic expression both globin chains, without testing knockdown of endogenous transcripts)
- (4) Demonstration of potent impact of HBB expression on both survival and blood-borne metastasis by tumor cells (including patient-derived CTCs), whereas the impact on primary tumor growth is minimal (this has never been demonstrated)
- (5) Clinical correlations with adverse prognosis, both in databases and in primary CTCs where beta-globin expression has not been previously been correlated with any outcome.

Based on these points, we are confident that our study provides considerable novelty and will reignite a field in which a few relatively limited observations had been published in the past. We have ensured that all referencing of these past publications are accurate.

Additional comments:

On Page 8, it is stated "Out of 21 candidates, only three genes, KLF4, KLF6 and ATF5, are induced when H727 cells are treated with hydrogen peroxide or grown under anchorage-free conditions (Figures S5B, C)." yet the figure S5B shows 6 genes are statistically induced by hydrogen peroxide. Please clarify.

We thank the reviewer for pointing out this confusion. Out of the 6 genes that are

induced by hydrogen peroxide, only KLF4, KLF6 and ATF5 are statistically induced under anchorage-free conditions (Figure S5C). We have now clarified this in the text.

On Page 6 "The specific overexpression of HBB, but not HBA, is also evident using single cell RNA-sequencing of CTCs and CTC-clusters from multiple independent breast and prostate cancer patients (Figures 1A,D,E)." However, only 1A includes HBA data.

We have edited the text accordingly.

On Page 12, "HBB mRNA level in primary tumor samples is generally low (0-0.1%), while increased expression is detected in 7 out of 19 metastatic prostate tumors (6.3-18.1%) (Figures 4E, S8)" However, the Figures referred to do not support this statement since 4E = localized 0.9%, met 15.9% and S8 shows RNA in situ.

We apologize and have corrected this typographic error.

On Page 12 "Expression of HBB is low in primary tumor cells, markedly induced in CTCs, but is sustained at an intermediate level in metastases (Figure S9A, B)." Is there a statistical difference between PT and Met in S9A?

The reviewer has a good point and we agree. Due to the heterogeneously bimodal distribution of Hbb expression in single metastatic tumor cells, the statistical difference is not significant. We have edited the text accordingly.

We thank all four reviewers for their helpful comments and hope that we have fully addressed their questions.

Sincerely,
Daniel Haber and Shyamala Maheswaran

REVIEWERS' COMMENTS:

Reviewer #1 (Remarks to the Author):

The revised manuscript has adequately addressed my concerns and the conclusion is strongly supported by the data.

Reviewer #2 (Remarks to the Author):

My previous points of criticism have been appropriately addressed and initially confusing aspects were removed. The manuscript addresses an important and timely issue which could have a substantial impact on the understanding (and potentially prevention) of metastasis formation in solid tumors. I do not have any concerns about the data quality or correctness. Therefore, I recommend the acceptance of this manuscript for publication in Nature Communications. I still have concerns about the general conclusions (which, however, should not preclude acceptance of the manuscript):

1. In the light of the incompletely understood contribution of CTCs to tumor metastasis formation, I am still not fully convinced about the global significance of the reported finding. My doubts are augmented by the enormous variance of HBB expression in the CTCs analyzed (figure 1A).
2. Some data still provide only circumstantial evidence: Most importantly, the authors have not formally shown that HBB is an important scavenger of ROS in the CTCs analyzed. They appear to agree with this in mentioning "Further biochemical studies will be required to fully characterize the physiological properties of beta-globin tetramers as ROS scavenger within CTCs." (p. 17)

Reviewer #3 (Remarks to the Author):

All my previous points have been addressed sufficiently.

Reviewer #4 (Remarks to the Author):

Reviewer 4 Response to the author's rebuttle>

It is clear now that we are now simply dealing with a matter of differing opinions (authors and this reviewer) concerning the relative novelty and suitability of the current manuscript for publication in Nature Communications and as such there is no definitive right or wrong. However, since novelty is a major issue, the author's statement that "...comments of the other reviewers and the responses that we have received to presentations of this work are consistent with it being a highly unexpected observation..." is at odds with the original review of Reviewer 1 which states "The study is well done and the overall conclusion appears to be sound. However, its significance is unclear, since the biological phenomenon revealed is not sufficiently novel, and the design of some experiments is confusing."

This study does report interesting findings including the ones listed by the authors in their rebuttal and of course the previous publications did not look at cancer cells upregulating beta-globin as a mechanism of dealing with ROS as they circulate through the bloodstream. Nevertheless, it is the opinion of this reviewer that both the original and revised are not sufficiently novel for publication in Nature Communications.

REVIEWERS' COMMENTS:

Reviewer #1 (Remarks to the Author):

The revised manuscript has adequately addressed my concerns and the conclusion is strongly supported by the data.

We thank the reviewer for his/her positive comments.

Reviewer #2 (Remarks to the Author):

My previous points of criticism have been appropriately addressed and initially confusing aspects were removed. The manuscript addresses an important and timely issue which could have a substantial impact on the understanding (and potentially prevention) of metastasis formation in solid tumors. I do not have any concerns about the data quality or correctness. Therefore, I recommend the acceptance of this manuscript for publication in Nature Communications.

We thank the reviewer for his/her positive comments.

I still have concerns about the general conclusions (which, however, should not preclude acceptance of the manuscript):

1. In the light of the incompletely understood contribution of CTCs to tumor metastasis formation, I am still not fully convinced about the global significance of the reported finding. My doubts are augmented by the enormous variance of HBB expression in the CTCs analyzed (figure 1A).

We understand the reviewer's concern. The variance of HBB expression presumably correlates with the considerable heterogeneity among cancer cells, especially among CTCs. We have previously published single-cell RNA sequencing data of prostate CTCs, demonstrating considerable heterogeneity, even among lineage-derived transcripts in CTCs from the same patient (Miyamoto, Zheng et al., Science 2015). We see comparable heterogeneity in HBB expression by CTCs.

2. Some data still provide only circumstantial evidence: Most importantly, the authors have not formally shown that HBB is an important scavenger of ROS in the CTCs analyzed. They appear to agree with this in mentioning "Further biochemical studies will be required to fully characterize the physiological properties of beta-globin tetramers as ROS scavenger within CTCs." (p. 17)

We agree with the reviewer's concern, and will further pursue this question in a more biochemical context in the future, as stated in our manuscript.

Reviewer #3 (Remarks to the Author):

All my previous points have been addressed sufficiently.

We thank the reviewer for his/her positive comments.

Reviewer #4 (Remarks to the Author):

Reviewer 4 Response to the author's rebuttle>

It is clear now that we are now simply dealing with a matter of differing opinions (authors and this reviewer) concerning the relative novelty and suitability of the current manuscript for publication in Nature Communications and as such there is no definitive right or wrong. However, since novelty is a major issue, the author's statement that "...comments of the other reviewers and the responses that we have received to presentations of this work are consistent with it being a highly unexpected observation...." is at odds with the original review of Reviewer 1 which states "The study is well done and the overall conclusion appears to be sound. However, its significance is unclear, since the biological phenomenon revealed is not sufficiently novel, and the design of some experiments is confusing."

This study does report interesting findings including the ones listed by the authors in their rebuttal and of course the previous publications did not look at cancer cells upregulating beta-globin as a mechanism of dealing with ROS as they circulate through the bloodstream. Nevertheless, it is the opinion of this reviewer that both the original and revised are not sufficiently novel for publication in Nature Communications.

We appreciate the reviewer's candor and respectfully disagree regarding the novelty of our findings.

We thank all four reviewers for their helpful comments.

Sincerely,
Daniel Haber and Shyamala Maheswaran